# Modulation of anti-tumor immunity by the brain's reward system

Tamar L Ben-Shaanan[1,2,3], Maya Schiller[1,2,3], Hilla Azulay-Debby[1,2,3], Ben Korin[1,2,3], Nadia Boshnak[1,2,3], Tamar Koren[1,2,3], Maria Krot[1,2,3], Jivan Shakya[1,4], Michal A. Rahat[1,4], Fahed Hakim[1,5,6] & Asya Rolls [1,2,3]

Regulating immunity is a leading target for cancer therapy. Here, we show that the anti-tumor immune response can be modulated by the brain's reward system, a key circuitry in emotional processes. Activation of the reward system in tumor-bearing mice (Lewis lung carcinoma (LLC) and B16 melanoma) using chemogenetics (DREADDs), resulted in reduced tumor weight. This effect was mediated via the sympathetic nervous system (SNS), manifested by an attenuated noradrenergic input to a major immunological site, the bone marrow. Myeloid derived suppressor cells (MDSCs), which develop in the bone marrow, became less immunosuppressive following reward system activation. By depleting or adoptively transferring the MDSCs, we demonstrated that these cells are both necessary and sufficient to mediate reward system effects on tumor growth. Given the central role of the reward system in positive emotions, these findings introduce a physiological mechanism whereby the patient's psychological state can impact anti-tumor immunity and cancer progression.

[1] Department of Immunology, Rappaport Faculty of Medicine, Technion - Israel Institute of Technology, 3525422 Haifa, Israel. [2] Department of Neuroscience, Rappaport Faculty of Medicine, Technion - Israel Institute of Technology, 3525422 Haifa, Israel. [3] The Technion Integrated Cancer Center, Technion - Israel Institute of Technology, 3525422 Haifa, Israel. [4] The Immunotherapy Lab, Carmel Medical Center, 3436212 Haifa, Israel. [5] Pediatric Pulmonary Unit, Rambam Health Care Campus, 3109601 Haifa, Israel. [6] Cancer Research Center, EMMS Hospital, 16100 Nazareth, Israel. These authors contributed equally: Tamar L Ben-Shaanan, Maya Schiller. These authors jointly supervised this work: Fahed Hakim, Asya Rolls. Correspondence and requests for materials should be addressed to F.H. (email: f_hakim@rambam.health.gov.il) or to A.R. (email: rolls@tx.technion.ac.il)

Epidemiological evidence supports a connection between the patient's mental state and cancer survival[1,2]. Nevertheless, many of these studies have yielded inconsistent results[3,4], and our understanding of the central neuronal mechanisms underlying the effect of emotional states on cancer is limited. Moreover, most research in this field has been focused on negative emotional states, such as stress and depression[5,6], while the impact of positive mental attributes on cancer biology is largely unknown.

The brain's reward system, specifically the dopaminergic neurons in the ventral tegmental area (VTA), constitutes a key neuronal network whose activation mediates positive emotions, expectations, and motivation[7–9]. The dopaminergic projections from the VTA to components of the limbic system are causally associated with motivated behavior and reward perception[10,11]. Pharmacological studies indicated a connection between reward system activity and immune modulation[12–14], and we recently showed that reward system activity can boost antibacterial immunity[15].

Immune system activity is crucial for controlling the initiation and progression of tumors. However, the immune system can also act as a double-edged sword. On the one hand, it generates effector cells, such as CD8 T cells and NK cells that can eliminate tumors[16,17]. On the other hand, some immune cell subsets, such as myeloid derived suppressor cells (MDSCs), act to support tumor growth by suppressing the anti-tumor immune response and by generating a favorable environment for the tumor (e.g., promoting angiogenesis)[18].

Thus, given the importance of the immune system in tumor biology, and since reward system activity affects immunity, we test here the hypothesis that reward system activity could affect tumor growth. We used chemogenetics, which enables targeted neuronal manipulation[19], to reveal a causal connection between reward system activity and alterations in anti-cancer immunity. We demonstrate, using two murine tumor models (Lewis lung carcinoma (LLC) and B16 melanoma), that chemogenetic activation of the reward system attenuates tumor growth. This manipulation also resulted in reduced sympathetic activity in the bone marrow, evident by attenuated noradrenaline (NA) levels. We further showed that MDSCs that develop in the bone marrow are functionally affected by the noradrenergic input. Thus, following reward system activation, MDSCs exhibited an attenuated immunosuppressive profile, which manifested in vivo by increased expression of Granzyme B by tumor CD8 T cells. By depleting and adoptively transferring MDSCs, we showed that these cells are both necessary and sufficient to mediate the effects of reward system activation on tumor growth.

## Results

### Specific and functional DREADD expression in the VTA.
Given the central role of the immune system in fighting cancer, and given the effects of reward system activity on immunity, we hypothesized that reward system activity could also affect tumor growth. To test this hypothesis, we used Designer Receptor Exclusively Activated by Designer Drugs (DREADDs) to specifically control reward system activity. DREADDs are mutated muscarinic receptors that no longer respond to their endogenous ligand[20]. Instead, upon exposure to a synthetic ligand, clozapine-N-oxide (CNO), stimulatory DREADDs (hM3D(Gq)) elicit an intracellular cascade that leads to neuronal activation[21]. DREADDs were expressed in VTA dopaminergic neurons, using an adeno associated virus (AAV)-based vector. The virus carried a gene encoding the DREADD receptor and an mCherry fluorescent reporter. We used stereotactic injections to deliver the virus directly to the VTA, and a Cre-dependent system to ensure virus expression specifically by the VTA dopaminergic neurons[22] (Fig. 1a). As controls in this study, we injected mice with the same virus encoding the fluorescent reporter, mCherry, but lacking the DREADD gene (control virus). This enabled us to control for any potential local inflammatory response induced by the viral infection, effects of surgery, and CNO administration.

The efficiency of virus-expression among the VTA dopaminergic neurons was validated using immunohistochemistry. We analyzed the co-labeling for tyrosine hydroxylase (TH), a marker for dopaminergic neurons[23,24], and mCherry, indicating virus expression (Fig. 1b). We found that $57.3 \pm 5\%$ of the $TH^+$ neurons in our control mice (injected with the control virus that lacks the information for DREADDs) and $64.7 \pm 3.4\%$ of $TH^+$ neurons in the experimental mice expressed the viral vector ($P = 0.29$; Supplementary Fig. 1; Student's $t$-test), rendering them potentially amenable for manipulation using DREADDs. We also confirmed that DREADDs expression was restricted to the VTA as demonstrated by the lack of mCherry labeling in additional brain regions (nucleus accumbens, lateral hypothalamus and frontal cortex) (Supplementary Fig. 2). To validate neuronal activation following daily CNO injection, we used immunohistochemical analysis of c-Fos, an early activation marker. Indeed, $60.9 \pm 4\%$ of the DREADD-expressing VTA neurons were c-Fos$^+$, compared to $34.5 \pm 6\%$ in the mice injected with the control virus ($P < 0.005$; Fig. 1c; Supplementary Fig. 3; Student's $t$-test).

### VTA activation attenuates tumor growth.
To analyze the effect of VTA activation on tumor growth, we used the solid tumor model LLC. We subcutaneously (s.c.) injected the LLC cells to the lower right flank of both DREADD-expressing mice and their controls (expressing the control virus). Following the tumor cell injection, we treated mice daily with CNO to induce VTA activation (controls were also treated with CNO; Fig. 1a). After 14 days of repeated VTA activation, tumor size was reduced by $46.5 \pm 17.6\%$ and tumor weight by $52.4 \pm 15.1\%$ ($P < 0.014$, $P < 0.003$, respectively) in the VTA-activated mice when compared to the controls (Fig. 1d–f). Reward system activation was also effective in reducing tumor size in an additional murine cancer model, the B16 melanoma model. As with the LLC model, we s.c. injected mice with B16 tumor cells and activated their reward system daily. In this cancer model, reward system activation reduced tumor weight by $38 \pm 11.99\%$ when compared to controls ($P < 0.03$; Fig. 1g), indicating that the effects of VTA activation were not limited to a single cancer model. Yet, as the effect was more profound with the LLC, we focused on this model for the remainder of the study. With this model, we tested an additional reward system activation regimen, every other day (instead of every day). To maintain the same number of VTA activation sessions, we continued this experiment for 28 days (instead of 14 days). As shown in Fig. 1h, this less frequent manipulation regimen, also reduced tumor weight by $28 \pm 11.7\%$ ($P < 0.006$). Taken together, the use of DREADDs, which enables causal connections to be established[25], revealed that VTA activation reduces tumor growth. However, it is unlikely that this is a direct effect, as dopamine does not cross the blood brain barrier (BBB), raising the question of how the signal is transmitted from the brain to the tumor.

### Necessity of sympathetic nervous system for VTA's effect on tumor.
Previous findings indicate that reward system activity affects the sympathetic nervous system (SNS)[15,26]. Other studies demonstrate that sympathetic activity affects cancer progression[27,28]. Thus, we hypothesized that the SNS is likely to be involved in mediating the signal from the VTA to affect tumor

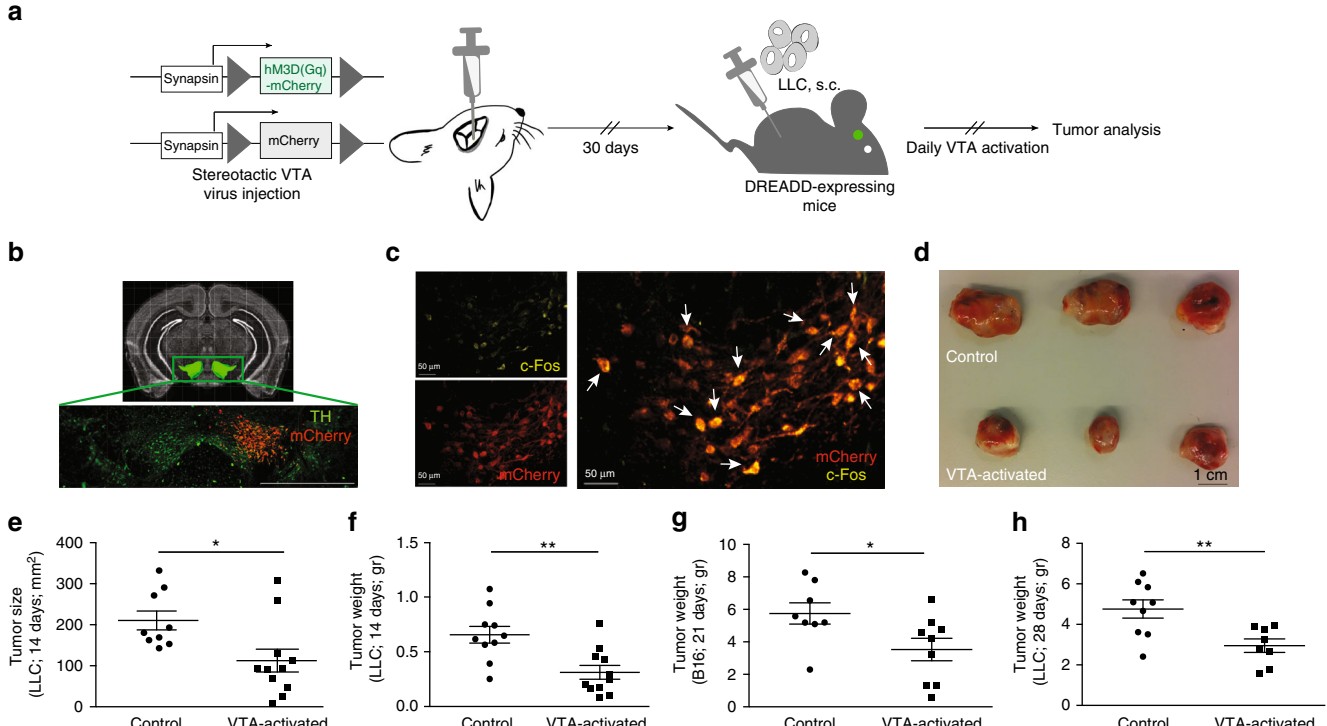

**Fig. 1** Repeated activation of DREADD-expressing neurons in the VTA reduces tumor size. **a** Schematic representation of the experimental design, which includes stereotactic injection of the virus to induce DREADD expression, injection of Lewis lung carcinoma tumor cells (LLC), repeated CNO injections for 14 days to activate the VTA dopaminergic neurons, and analysis of tumor size. For all in vivo experiments, the control mice were injected with the virus carrying the gene encoding mCherry but lacking the information encoding for DREADDs. These controls were treated, as the experimental group, with daily CNO injections. **b** Demonstration of localization and specificity of DREADD expression in the VTA. Upper panel: VTA localization based on the Allen Brain Atlas. Lower panel: Immunohistochemical demonstration of virus localization in the VTA indicated by mCherry staining (red) co-localized with TH expression (green) in the right VTA dopaminergic neurons (virus was injected unilaterally to the right hemisphere; scale bar, 500 μm). **c** c-Fos expression, indicating cell activation co-localized with DREADD expression in VTA neurons after 14 days of repeated CNO injections to tumor-bearing mice (scale bar, 50 μm). **d** Representative tumors dissected from control mice (injected with virus carrying the gene encoding mCherry but not the DREADD gene), and VTA-activated mice. Both groups were subjected to daily CNO injection for 14 days. **e** Size of LLC tumors following sacrifice in mice subjected to daily VTA activation for 14 days, and their controls ($P < 0.014$; Student's $t$-test; mean ± s.e.m; $n = 9, 11$). Data represent two independent repeats. **f** Weight of LLC tumors following sacrifice in mice subjected to daily VTA activation for 14 days, and their controls ($P < 0.003$; Student's $t$-test; mean ± s.e.m; $n = 10, 11$). Data represent two independent repeats. **g** Weight of B16 tumors following sacrifice in mice subjected to daily VTA activation for 21 days, and their controls ($P < 0.03$; Student's $t$-test; mean ± s.e.m; $n = 8, 9$). Data represent two independent repeats. **h** Weight of LLC tumors following sacrifice in mice subjected to VTA activation every other day for 28 days, and their controls ($P < 0.006$; Student's $t$-test; mean ± s.e.m; $n = 9, 8$). Data represent two independent repeats

growth. To directly test this hypothesis, we treated DREADD-expressing mice and their controls with 6-hydroxydopamine (6OHDA; intraperitoneal injection). 6OHDA systemically ablates catecholaminergic neurons comprising the SNS[29]. 6OHDA does not cross the BBB[29], and thus, when injected to the periphery, its effect is considered to be limited to the peripheral SNS innervations. Accordingly, 6OHDA had no effect on the number of TH+ cells in the VTA of the tumor-bearing mice ($P < 0.69$; Supplementary Fig. 4). In these sympathetically ablated mice, VTA activation had no effect on tumor weight ($P < 0.8825$; Fig. 2a) demonstrating the requirement of the SNS in mediating VTA effects on the tumor. We further validated the SNS involvement in mediating reward system effects by treating the VTA-activated mice and controls with a blocker to one of the main NA receptors, the β-adrenergic receptor. Nadolol is a β-adrenergic blocker that does not cross the BBB[30] and thus, inhibits β-adrenergic receptors in the periphery. Similarly to 6OHDA treatment, Nadolol eliminated the effects of VTA-activation on tumor weight ($P < 0.4283$; Fig. 2b). Thus, we conclude that sympathetic activity is required to mediate the reward system effects on tumor growth.

SNS activity is commonly associated with the stress response. Therefore, we analyzed the levels of the stress hormone, corticosterone, in VTA-activated mice and their controls. We could not detect any difference in plasma corticosterone between the two groups ($P < 0.81$; Fig. 2c). However, the SNS also directly innervates organs relevant to cancer biology, including some tumors[27,31] and all immune organs[32,33]. These innervations enable local control of sympathetic activity at the target site[34,35]. Thus, we analyzed whether VTA activation had any effect on these local sympathetic innervations, measuring the levels of the primary SNS neurotransmitter, NA, in various target sites. VTA activation had no significant effect on plasma and spleen NA levels (determined by ELISA; $P < 0.89$, $P < 0.16$; Fig. 2d, e; Supplementary Fig. 5), though at the tumor site, we observed some, non-significant ($P < 0.107$) reductions in NA level (Fig. 2f; Supplementary Fig. 5). However, in the bone marrow of VTA-activated mice, NA levels were significantly reduced by 24.6 ± 0.1% ($P < 0.044$; Fig. 2g; Supplementary Fig. 5). We verified this finding by staining the bone marrow for TH, an enzyme that participates in NA synthesis and is expressed by sympathetic fibers in the periphery[36]. We found a 50.5 ± 22.3% reduction in

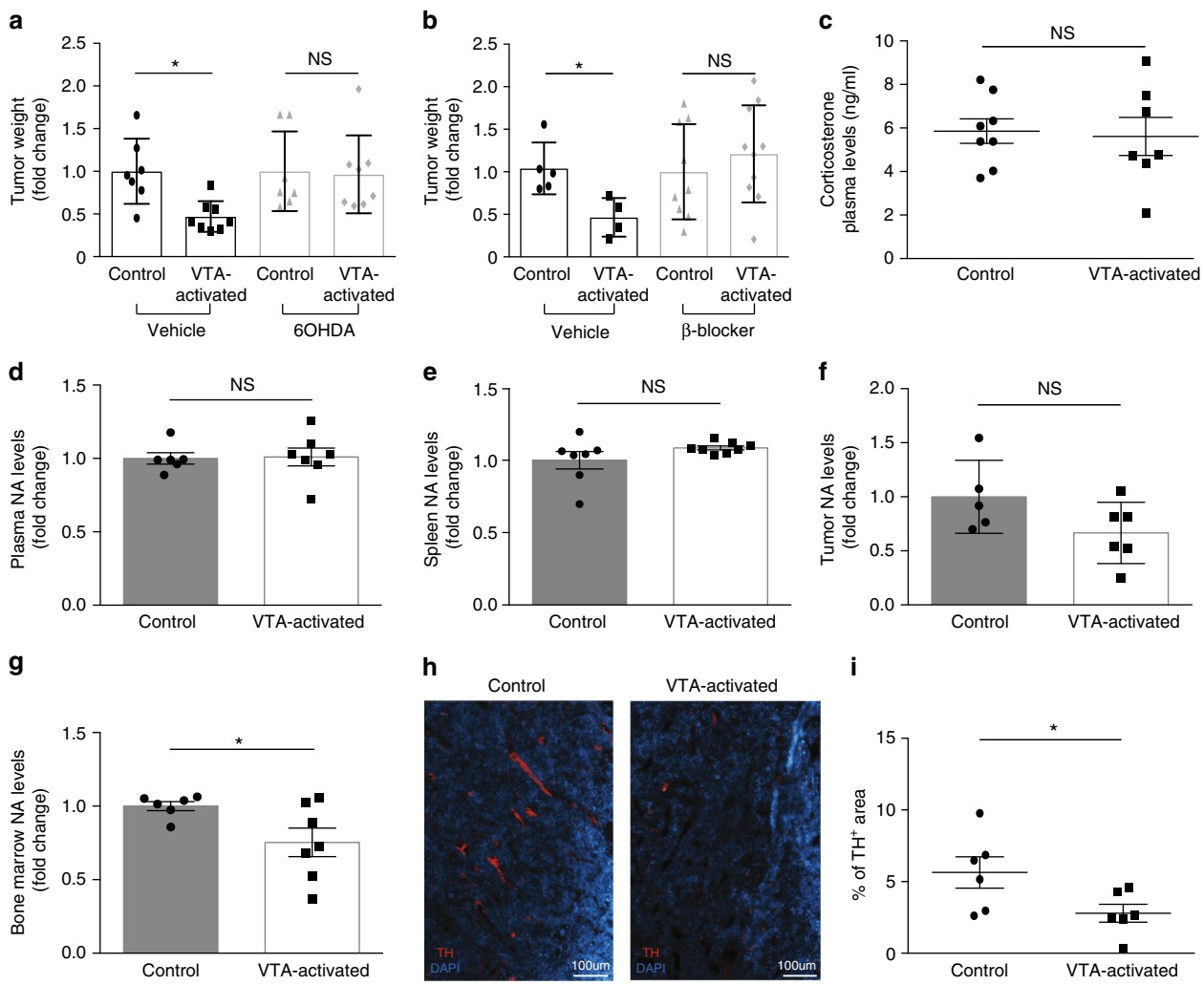

**Fig. 2** Peripheral SNS activity is necessary for the effect of VTA activation on tumor growth. **a** Four groups of mice were included: two groups expressing DREADDs and two controls (injected with virus lacking the DREADD gene). Each of them was then treated with 6OHDA or vehicle. All four groups were injected with LLC cells and treated with CNO. After 14 days of VTA activation, the weight of the tumors was analyzed ($P < 0.8825$; Student's $t$-test; mean ± s.e.m; NS not significant; $n = 7, 8$ in each group). Data are shown as fold change in tumor weights relative to the mean in the control group. Data represent two independent repeats. **b** Weights of tumors isolated from DREADDs-expressing mice and their controls (injected with virus lacking the DREADD gene) that were treated daily with CNO and injected (intraperitoneal) daily with Nadolol, or a vehicle starting the day of tumor cell injection ($P < 0.4283$; Student's $t$-test; mean ± s.e.m; NS not significant; $n = 5, 4$ for vehicle and $n = 9, 10$ for β-blocker experiment). Data are shown as fold change in tumor weights relative to the mean of the control. Data represent two independent repeats. **c** Plasma samples of tumor-bearing, VTA-activated mice and their controls. Samples were analyzed for their corticosterone levels using a standard ELISA kit ($P < 0.81$; Student's $t$-test; mean ± s.e.m; $n = 8, 7$). Data represent two independent repeats. **d** Noradrenaline levels measured by ELISA in the plasma ($P < 0.89$; Student's $t$-test; mean ± s.e.m; $n = 6, 7$), **e** spleen ($P < 0.16$; Student's $t$-test; mean ± s.e.m; $n = 7, 8$), **f** tumor ($P < 0.107$; Student's $t$-test; mean ± s.e.m; $n = 5, 6$), and **g** bone marrow ($P < 0.044$; Student's $t$-test; mean ± s.e.m; $n = 6, 7$) of tumor-bearing VTA-activated mice and controls (control virus+CNO). Data are represented as fold change between VTA-activated mice and controls, from two independent repeats. **h** Immunohistochemical staining for tyrosine hydroxylase (TH; red) in the bone marrow of tumor-bearing VTA-activated mice and controls (control virus+CNO) following 14 days of repeated CNO injections (blue: DAPI staining; scale bar, 100 μm). **i** Analysis of the percentage of TH-positive areas per field of view ($P < 0.047$; Student's $t$-test; mean ± s.e.m; $n = 6$). Data represent two independent repeats

TH expression in the bone marrow of tumor-bearing VTA-activated mice compared to their controls ($P < 0.044$; Fig. 2h, i).

**Functional β2-adrenergic receptor on bone marrow MDSCs.** The bone marrow is a particularly important site for the anti-tumor immune response[37,38]. During cancer progression, the bone marrow is characterized by extensive proliferation of myeloid cells, especially of MDSCs (identified by Gr-1+ CD11b+ expression)[39,40]. In the LLC model, this heterogeneous population, including neutrophils[41], comprises a major fraction of cells

in the bone marrow of tumor-bearing mice[39,42] (on average 81.6 ± 1.6% of the bone marrow cell population 28 days after s.c. injection of LLC; Fig. 3a). MDSCs support tumor progression by various mechanisms including promoting angiogenesis and suppressing anti-tumor immunity[42,43]. Given that VTA activation specifically reduced NA levels in the bone marrow and given the high abundance of MDSCs in this site, we decided to focus on these cells. Moreover, previous studies showed that SNS activity affects myeloid population in the bone marrow[44–47].

First, we evaluated whether MDSCs could be affected by the change in NA levels, and analyzed their expression of the β2

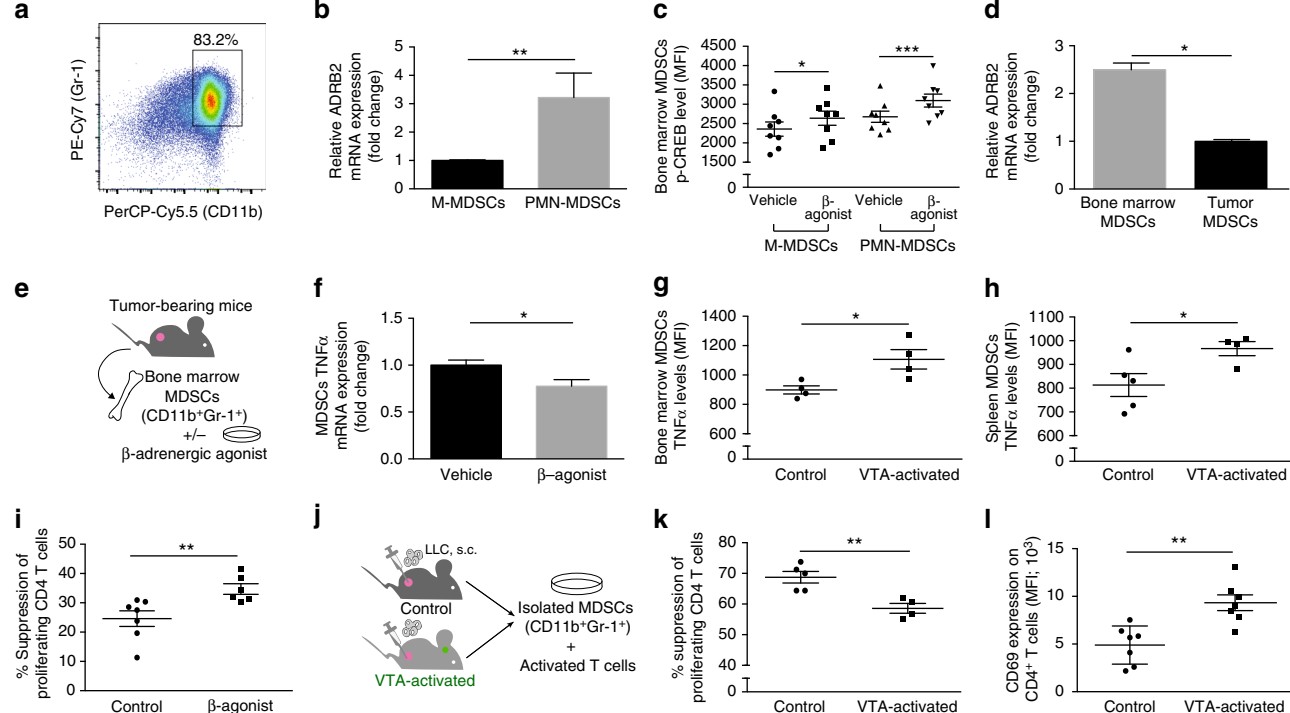

**Fig. 3** Bone marrow MDSCs express a functional β2-adrenergic receptor and are affected by VTA activation. **a** Representative dot plot demonstrating staining for Gr-1 and CD11b in the bone marrow of LLC tumor-bearing mice. **b** qPCR analysis of β2-adrenergic receptor (ADRB2) mRNA expression levels by bone marrow M-MDSCs and PMN-MDSCs of tumor-bearing mice ($P < 0.004$; Mann–Whitney test; $n = 5$, 6 in each group). **c** Phospho-flow analysis of intracellular CREB phosphorylation levels in bone marrow MDSCs following incubation with isoproterenol (1 μM; 15 min). Values represent median florescence intensity (MFI) (M-MDSCs $P < 0.023$, PMN-MDSCs $P < 0.0004$; Student's paired $t$-test; mean ± s.e.m; $n = 8$). Data represent two independent repeats. **d** qPCR analysis of ADRB2 mRNA expression levels by MDSCs sorted from the bone marrow or tumor of LLC tumor-bearing mice ($P < 0.015$; Mann–Whitney test; $n = 5$). Data represent two independent repeats. **e** MDSCs were isolated from the bone marrow of tumor-bearing mice and incubated with isoproterenol (1 μM). Changes in gene expression were analyzed by qPCR. **f** qPCR analysis of TNFα mRNA expression levels by bone marrow MDSCs of tumor-bearing mice, and incubated with isoproterenol (1 μM) ($P < 0.016$; Mann–Whitney test; $n = 5$). Data represent one experiment out of two independent repeats; *$P < 0.1$. **g** Intracellular TNFα levels in bone marrow MDSCs from VTA-activated mice and controls (injected with virus lacking the DREADD gene). Values represent MFI ($P < 0.027$; Student's $t$-test; mean ± s.e.m; $n = 4$). **h** Intracellular TNFα levels in spleen MDSCs from VTA-activated mice and controls (injected with virus lacking the DREADD gene). Values represent MFI ($P < 0.039$; Student's $t$-test; mean ± s.e.m; $n = 5$, 4). **i** Analysis of suppression assay (as described in the methods) using bone marrow MDSCs from tumor-bearing mice following treatment with isoproterenol or vehicle ($P < 0.007$; Student's $t$-test; mean ± s.e.m; $n = 7$, 6). Data represent two independent repeats. **j** Schematic representation of the experiment measuring the effect of VTA activation on MDSCs immunosuppressive capacity. **k** Analysis of suppression assay using tumor MDSCs from VTA-activated mice and controls (injected with virus lacking the DREADD gene; $P < 0.004$; Student's $t$-test; mean ± s.e.m; $n = 5$, 4). Data represent two independent repeats. **l** CD69 expression levels on tumor CD4$^+$ T cells from VTA-activated mice and controls, indicated by MFI ($P < 0.001$; Student's $t$-test; mean ± s.e.m; $n = 7$). Data represent two independent repeats

adrenergic receptor. We focused on this receptor because it was already shown that it is commonly expressed on immune cells[48], and it was consistent with our finding that the β-blocker, Nadolol, abrogated VTA-activation effects on tumor growth (Fig. 2b). We compared β2 adrenergic receptor mRNA levels expressed by two main subsets of bone marrow MDSCs: monocytic MDSCs (M-MDSCs; CD11b$^+$ Ly6C$^+$) and polymorphonuclear MDSCs (PMN-MDSCs; CD11b$^+$ Ly6G$^+$). While both MDSCs subsets expressed the β2 adrenergic receptor, receptor levels on PMN-MDSCs were 3.2-fold higher compared to M-MDSCs ($P < 0.004$; Fig. 3b). Next, we tested whether the β2 adrenergic receptor was not only expressed by these cells, but also functionally active. NA signaling is known to activate an intracellular cascade leading to CREB phosphorylation[49], thus to evaluate the activity of the receptor, we assessed pCREB levels. We exposed PMN-MDSCs and M-MDSCs to a β-adrenergic agonist (isoproterenol) in vitro. PMN-MDSCs manifested a significantly greater increase in pCREB levels compared to M-MDSCs following exposure to the β-agonist ($P < 0.0004$, $P < 0.023$, respectively; Fig. 3c; Supplementary Fig. 6), in agreement with their elevated expression of

the β2 adrenergic receptor (Fig. 3b). Although PMN-MDSCs were more reactive than M-MDSCs to NA signaling, both populations responded to this signal. Moreover, VTA activation had no effect on the relative abundance of the two subpopulations ($P < 0.22$ for PMN-MDSCs and $P < 0.64$ for M-MDSCs; Supplementary Fig. 7); therefore, we continued our analysis on the total MDSC population. We found that the overall expression levels of the β2 adrenergic receptor on MDSCs isolated from the bone marrow were 2.49-fold higher compared to MDSCs isolated from the tumor ($P < 0.015$; Fig. 3d), suggesting that bone marrow MDSCs are more susceptible to changes in NA levels. Interestingly, VTA activation reduced NA levels mainly in the bone marrow (Fig. 2g–i).

To determine the direct effects of NA on MDSCs, we isolated these cells from the bone marrow of tumor-bearing mice and incubated them in vitro with the β-adrenergic agonist (Fig. 3e). We screened for changes in the expression of known mediators that can affect tumor growth. Specifically, we analyzed mRNA levels of the angiogenic factor VEGF[42], the pro-tumorigenic cytokine TGFβ[50], the immunosuppressive cytokine IL-10 as well

as the regulatory factors iNOS and TNF-α[50–52]. This analysis revealed that following β-adrenergic agonist treatment, the main significant effect was a reduction in TNF-α expression levels by MDSCs ($P < 0.016$; Fig. 3f; Supplementary Fig. 8). TNF-α is a potent anti-tumor cytokine that has a disputed, yet central role in cancer biology[53]. Given that the β-adrenergic agonist decreased TNF-α expression in the bone marrow MDSCs, we expected that in the VTA-activated mice, which had reduced NA levels, MDSCs would manifest an increase in TNF-α. Accordingly, flow cytometry analysis revealed a significant increase in TNF-α levels in MDSCs from the bone marrow and spleen ($P < 0.027$, $P < 0.039$, respectively; Fig. 3g, h). This increase was specific to TNF-α, as other markers such as IFN-γ, iNOS, Arginase, IDO, PDL-1, and VEGF were not affected ($P < 0.98$, $P < 0.44$, $P < 0.34$, $P < 0.35$, $P < 0.36$, and $P < 0.95$, respectively; Supplementary Fig. 9). Moreover, CD31 mRNA levels in the tumor site were unchanged following VTA activation ($P < 0.41$; Supplementary Fig. 10). . Taken together, following VTA activation, we observed a decrease in NA levels in the bone marrow but not in the spleen or tumor site. Thus, it is possible that this change in the bone marrow, the milieu, where the MDSCs develop, affected their subsequent functional profile at other sites.

One of the pro-tumorigenic functions of MDSCs is their ability to suppress T-cell activation and proliferation. Therefore, we treated MDSCs with the β-adrenergic agonist and incubated them with activated T cells in vitro. We found that the β-adrenergic agonist increased the immunosuppressive effect of MDSCs, manifested by a reduction in CD4 T-cells proliferation ($P < 0.007$; Fig. 3i). In agreement with this in vitro finding, MDSCs isolated from tumors of VTA-activated mice were less effective in suppressing CD4 T cell proliferation compared to MDSCs derived from controls ($P < 0.004$; Fig. 3j, k; Supplementary Fig. 11). Moreover, CD4 T cells derived from the tumor site of VTA-activated mice manifested elevated levels of the activation marker, CD69 ($P < 0.001$; Fig. 3l).

**Necessity and sufficiency of MDSCs on VTA's effect.** Although CD69 is indicative of cell activation, it is not directly associated with effector and cytotoxic functions. Therefore, we characterized CD4 and CD8 populations in the spleen and tumor site. We evaluated their abundance and expression of the cytokines IFN-γ, TNF-α, and Granzyme B (on CD8 cells) (Supplementary Fig. 12). We found a significant increase in Granzyme B expression on CD8 T cells in the VTA-activated mice ($P < 0.0159$; Fig. 4a; Supplementary Fig. 13). Granzyme B is known to have a cytotoxic effects on target cells[54] and thus, the increase in its levels further supports the induction of anti-tumor immune response following VTA activation. To determine whether this effect on Granzyme B levels was dependent on MDSCs activity, we depleted the MDSC population. We injected an anti-Gr-1 antibody to deplete the Gr-1-positive populations in tumor-bearing VTA-activated mice and controls. As shown in Fig. 4a, depletion of MDSCs eliminated the effect of VTA activation on Granzyme B levels on CD8 T cells ($P < 0.1188$). Moreover, we found that the tumor weight in these anti-Gr-1-treated mice did not differ between the two experimental groups, indicating the necessity of MDSCs in mediating these effects ($P < 0.3272$; Fig. 4b).

Although we do not assume that MDSCs are the only cell population affected by VTA activation, a change in these cells might be sufficient to induce the reduction in tumor weight. To directly address this question, we performed an adoptive transfer experiment; MDSCs were isolated (based on their expression of Gr-1 and CD11b) from tumors of VTA-activated mice and their controls (carrying the control virus and treated with CNO). The cells were then co-injected along with new tumor cells into naïve

recipient mice. As the recipient mice were injected with an equal number of LLC cells and MDSCs (isolated from control or VTA-activated mice), any difference in the evolving tumors could be directly attributed to functional changes in the transferred cells (Fig. 4c). After validation that the transferred MDSCs could survive in the recipient mice (Supplementary Fig. 14) and that the tumor cells were not immediately rejected in our experimental paradigm ($P < 0.24$; Supplementary Fig. 15), we compared the tumors in both groups. We found that when MDSCs were derived from VTA-activated mice, the tumor size was reduced by 42.9 ± 13.4% and tumor weight was reduced by 43.6 ± 16.3% compared to the controls ($P < 0.008$, $P < 0.019$, respectively; Fig. 4d–f), indicating that the effect on MDSCs was sufficient to mediate the VTA effects on tumor growth. Taken together, the transfer and depletion experiments, demonstrated that MDSCs are both sufficient and necessary to mediate VTA-induced effects on tumor growth.

## Discussion

Taken together, in this study, we establish a causal link between reward system manipulation and tumor growth. We demonstrate that although the effect is causal, the link is indirect. The mechanism is dependent on SNS activity, specifically β-adrenergic input, and is mediated, at least in part, via MDSCs. It should be noted that MDSCs are a heterogeneous population, and the markers used for their characterization do not allow distinguishing between MDSCs and neutrophils[41]. Moreover, it is possible that additional effects induced by VTA activation are independent of the effects on MDSCs. The observed decrease in NA levels in the bone marrow following VTA activation is consistent with previous studies indicating that VTA activity and positive emotional state affect the SNS[15,55]. Moreover, the differential control of sympathetic activity at specific organs was described before[47,56], though the mechanisms underlying such specificity are, as yet, unknown.

Our analysis was focused on the effects of VTA activation on the anti-tumor immune response, specifically MDSCs. However, it is likely that such a brain manipulation will also affect other physiological systems, which can also contribute to the observed effects. It is important to note that we do not consider such robust and specific modulations of VTA activity (as induced here by DREADD activation) to be physiological and the relevance to human cancer is still unknown. Moreover, in this study, we used two tumor models that evaluate a specific aspect in tumor biology, namely, primary tumor growth rates. Thus, other aspects of cancer development and progression such as tumor initiation or metastasis remain unknown. Thus, rather than dissecting the specific molecular mechanisms mediating these effects, our study aims to demonstrate a functional connection between a mood-regulating neuronal circuit, such as the reward system, and tumor growth.

Moreover, given the central role of the reward system in mood regulation, the finding that the reward system can affect anti-tumor immunity and tumor growth introduces a new mechanistic insight into the epidemiological connection between mental states and cancer progression.

## Methods

**Mice.** Adult male (10–12 weeks of age; 20–25 g) TH-Cre mice (B6129X1-Th<tm1 (cre)Te>/Kieg; EMMA), GFP mice, and C57BL/6 mice were used in all experiments. Mice were maintained under Specific-Pathogen-Free (SPF) conditions on a 12:12 h light cycle (lights on at 07:00). All experiments were performed in accordance with the National Institutes of Health Guide for the Care and Use of Laboratory Animals. All procedures and protocols were approved by the Technion Administrative Panel of Laboratory Animal Care.

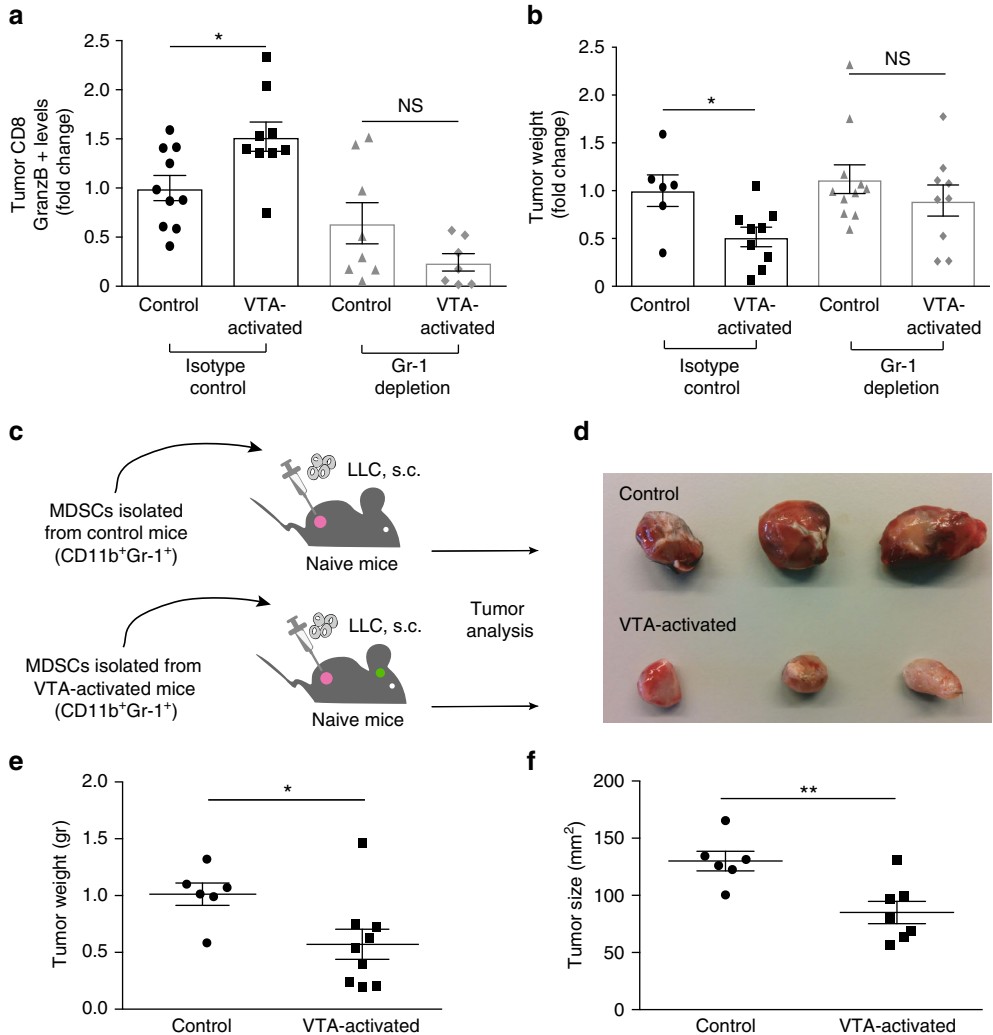

**Fig. 4** MDSCs are necessary and sufficient to mediate the effects of VTA activation on tumor weight. For all experiments in vivo, the control mice were injected with the control virus carrying the gene encoding mCherry but lacking the information for DREADD. These controls were treated, as the experimental group, with daily CNO injections. **a**, **b** Mice were daily injected (starting the day of tumor cell injection) with an anti-Gr-1 depletion antibody or an isotype control antibody, and with CNO to daily activate their VTA for 14 days. **a** Analysis of intracellular Granzyme-B levels in tumor CD8 T cells from VTA-activated mice and their controls. Data are shown as fold change in Granzyme B antibody staining (Gr-1 depletion ($P < 0.1188$; $n = 8, 7$), isotype control ($P < 0.0159$; $n = 10, 9$); Student's $t$-test; mean ± s.e.m; NS not significant). Data represent two independent repeats. **b** Weights of tumors isolated from DREADDs-expressing mice and their controls. Data are shown as fold change in tumor weight (Gr-1 depletion ($P < 0.3272$; $n = 11, 9$), isotype control ($P < 0.0202$; $n = 6, 9$); Student's $t$-test; mean ± s.e.m; NS not significant). Data represent two independent repeats. **c** Schematic representation of the experimental design used to determine whether the difference in MDSCs function induced by VTA activation is sufficient to recapitulate the effect of the VTA on tumor weight. **d** Representative tumors extracted from mice inoculated with naive LLC cells together with MDSCs isolated from mice that expressed the control virus (treated daily with CNO as the experimental group; upper row), versus LLC tumors inoculated with MDSCs isolated from VTA-activated animals (lower row). **e** Weights and **f** size of tumors extracted from mice inoculated with naive LLC cells together with MDSCs isolated from control animals (injected with virus lacking the DREADD gene), or isolated from VTA-activated mice (tumor weights $P < 0.019$, tumor size $P < 0.008$; Student's $t$-test; mean ± s.e.m; $n = 6, 9$). Data represent two independent repeats

**Stereotactic injection**. As described before[15], mice were anesthetized using a ketamine/xylazine mixture (ketamine 80 mg/kg; xylazine 15–20 mg/kg) diluted in sterile saline solution (NaCl 0.9%), before being fixed in a stereotactic frame (Stoelting, Wood Dale, IL, USA). Mice were randomly divided between VTA-activated and control group. An AAV8 virus, (AAV8-hSyn-DIO-hM3D(Gq)-mCherry) purchased from the Vector Core at the University of North Carolina, was used for Cre-dependent DREADD expression. We injected 0.7 µl of the virus ($10^{12}$–$10^{13}$ vg/ml) into the right VTA region (anterior–posterior −3.2 mm; medial–lateral 0.48 mm; dorsal–ventral 4.7 mm) of TH-Cre mice. Control mice were injected with a sham AAV8-hSyn-DIO-mCherry construct, lacking the DREADD gene. This allowed us to control for local inflammatory response, induced by the virus and CNO injection, and surgical procedure. Experiments were performed 30 days after surgery to ensure the expression of the DREADD. Mice showing signs of physical distress and pain were excluded from the experiment. Following sacrifice, stereotactic injection sites were verified by immunohistochemistry.

**Cell lines and culture**. The LLC and B16 melanoma cell lines were provided by the research groups of professors Yuval Shaked and Yishai Ofran of the Technion's Ruth and Bruce Rappaport Faculty of Medicine. The cells were originally obtained from ATCC. The LLC cell line was established from the lung of a C57BL mouse bearing a tumor. B16 melanoma is a common cell line that naturally originated in melanin-producing epithelia of C57BL mice. The cells were cultured in Dulbecco's modified Eagle's medium (DMEM; Biological Industries, IL) supplemented with 10% fetal bovine serum (FBS; Biological Industries, IL), 2 mM L-glutamine (Biological Industries, IL), 1 mM sodium pyruvate (Biological Industries, IL), and 1% PenStrep (Biological Industries, IL). The cells have been tested and found negative for mycoplasma. Cells were cultured in a humidified atmosphere of 95% air and 5% $CO_2$ at 37 °C.

**DREADD-induced VTA activation in tumor-bearing mice**. DREADD-expressing mice and their controls were injected s.c. with LLC or B16 cells ($0.5 \times 10^6$ cells/

mouse; $0.25 \times 10^6$ cells/mouse, respectively) by a researcher blinded to the experiment conditions. The cells from both lines were injected to the lower right flank of the mice in a total volume of 250 μl. The mice were then given a daily intraperitoneal injection of CNO (Sigma) diluted 1 mg/kg in saline. Control mice were similarly injected with CNO.

**Tissue preparation and immunohistochemistry**. Validation of the virus injection site, evaluation of DREADDs expression, c-Fos analysis, and assessment of $TH^+$ neurons in the bone marrow were performed by immunohistochemistry. Mice were sacrificed, and their brains were fixed in 4% paraformaldehyde (PFA) in PBS for 48 h, cryoprotected in 30% sucrose solution for another 48 h, and then frozen in dry ice. Coronal cryosections from the midbrain were sliced at 12 μm thickness and mounted on super-frost slides (Fisherbrand). The tissues were stained for TH with mouse anti-TH (1:200; Millipore, Billerica, MA, USA), and the proportion of DREADD-expressing cells ($mCherry^+$) out of the total $TH^+$ population in the VTA was evaluated (min 1500 cells). To evaluate c-Fos expression, the mice were sacrificed 90 min after CNO injection, and treated as described above. The fixed sections were prepared and stained with rabbit anti-c-Fos antibodies (1:100, Calbiochem, San Diego, CA, USA). The proportion of c-Fos$^+$ cell nuclei was calculated from the total number of virus-expressing cells. Quantification of double-positive cells was performed using Imaris Software. For the evaluation of the $TH^+$ neurons in the bone marrow, the mice were sacrificed, and the hind legs were fixed in 4% PFA in PBS for 48 h, and then decalcified in 0.5 M EDTA in PBS for 10 days. Afterwards, the bones were cryoprotected in 30% sucrose solution for another 48 h and then frozen in dry ice. Cryosections from the bone were sliced at 10 μm thickness and mounted on super-frost slides (Fisherbrand). The tissues were stained for TH with mouse anti-TH (1:200; Millipore, Billerica, MA, USA) and the TH positive neurons were quantified using Fiji software (expressed as area percentage). All images were taken at 20× or 10× magnification using an Axio imager M2 microscope (Carl Zeiss Inc. US).

**Analysis of NA and corticosterone levels**. For the analysis of NA and corticosterone levels in plasma samples, whole blood from anesthetized mice were collected in heparinized tubes. Samples were centrifuged for 15 min at $2000 \times g$ and stored at $-80\,°C$ until analyzed. To analyze NA levels in the bone marrow, mouse femurs and tibias were flushed with 2 ml of cold PBS and centrifuged for 10 min at $300 \times g$. Supernatants were collected and stored at $-80\,°C$. To analyze the NA levels in the spleen and tumor, 0.7 g of each sample was homogenized using Bullet Blender (Next Advance, US) in 700 μl of PBS containing 0.01 N HCl, 1 mM EDTA, and 4 mM sodium metabisulfite. NA levels in the plasma, spleen, tumor, and bone marrow sample were analyzed with Norepinephrine ELISA kit (IBL-America, US). Plasma corticosterone levels were analyzed with Corticosterone ELISA kit (Enzo Life Sciences, US).

**Flow cytometry**. Mice were sacrificed, and their bone marrows, spleens, and tumors were collected. Spleens were dissociated into single-cell suspensions in 2% FBS in PBS (Biological Industries, IL) and mesh-filtered to remove clumps and debris. Tumor tissues were dissociated into single-cell suspensions in 2% FBS (Biological Industries, IL) and 1 mM EDTA in PBS, and mesh-filtered to remove clumps and debris. Bone marrow cells were obtained by flushing mouse femurs and tibias with PBS. Cell suspensions were treated with red blood cell lysis buffer (BD Biosciences, NJ, USA). For extracellular staining, cells ($10^6$) were incubated with antibodies for 30 min at 4 °C. Then, cells were washed with FACS staining buffer (PBS containing 1% bovine serum albumin and 0.05% sodium azide). The following mAbs (all from Biolegend, San Diego, CA, US) were used: PerCP-conjugated anti-CD11b (M1/70), PE-Cy7 or PE-conjugated anti-Ly-6G/Ly-6C (Gr-1;RB6-8C5), PE or Alexa Flour 700-conjugated anti-CD45 (30-F11), Brilliant Violet 510-conjugated anti-Ly6C (HK1.4), FITC or APC-conjugated anti-Ly6G (1A8), PE-conjugated anti-TNF-α (MP6-XT22), APC-conjugated anti-Arginase (R&D Systems), APC-conjugated anti-PDL-1 (10F.9G2), Alexa Flour 647-conjugated anti-IDO-1 (2E2/IDO1), and Alexa Flour 405-conjugated anti-iNOS (Santa Cruz). For intracellular staining, the samples were first stained extracellular as described above, fixated and permeabilized with BD Cytofix/Cytoperm kit, and stained with the intracellular antibodies. The samples were re-suspended in 400 μl of 1% PFA and analyzed by flow cytometry. All samples were analyzed using an LSRFortessa cell analyzer and FlowJo software.

**Sympathetic denervation**. As described before[15], virus-injected mice (30 days after stereotactic injection) were sympathetically denervated by two intraperitoneal injections of 6OHDA (150 mg/kg in 0.01% ascorbic acid in saline; Sigma–Aldrich, St. Louis, MO, USA) administered at 24 h intervals. Mice were injected s.c. with LLC tumor cells in the lower right flank ($0.5 \times 10^6$ cells/mouse) 5 days after the last 6OHDA injection. The mice were injected daily with the DREADD ligand, CNO, for 14 days, and afterwards the immune organs and tumors were removed for analysis.

**Nadolol and anti Gr-1 treatment**. Virus-injected mice (30 days after stereotactic injection) were injected with tumor cells. Then, we injected ip either Nadolol (6 mg/kg in saline; Sigma–Aldrich, St. Louis, MO, USA) or anti-Gr-1 antibody (200

μg/mouse in saline; BioXcell) or an isotype control antibody (200 μg/kg in saline; BioXcell). The anti Gr-1 antibody, its isotype control as well as the Nadolol and its vehicle control, were all injected daily 20 min before the CNO injections. After 14 days, the mice were sacrificed and their tumors were analyzed.

**p-CREB analysis**. Cells from the bone marrow of tumor bearing mice were extracted by flushing femurs and tibias with 2 ml cold PBS. Then, the cells were centrifuged for 10 min at $300 \times g$, and the pellets were re-suspended at a concentration of $5 \times 10^6$/ml in RPMI 1640 medium (Biological Industries IL) supplemented with heat inactivated 10% fetal calf serum (Biological Industries IL), L-glutamine (200 mM; Biological Industries IL), sodium pyruvate (100 nM; Biological Industries IL), and Penicillin–Streptomycin solution (diluted 1:100; Biological Industries IL). Bone marrow cells extracted from each mouse were divided into two 24-well plates. Into one of the wells, the β-adrenergic agonist, isoproterenol, was added at a concentration of 1 μM for 15 min. After that, the cells were fixed with 4% PFA (ThermoFisher scientific) for 15 min, and then centrifuged for 5 min at $300 \times g$. The supernatants were discarded and the cell pellet was re-suspended in 1 ml 90% cold methanol for 20 min at 4 °C. Then, the cells were washed twice with FACS staining buffer, and stained with PerCP-conjugated anti-CD11b, Brilliant Violet 421-conjugated anti-Ly6C, PE-Cy7-conjugated anti-Ly-6G/Ly-6C (Gr-1), and Alexa Flour 488-conjugated anti-phospho-CREB (Ser133; from Cell Signaling Technology) for 1 h at room temperature. All samples were analyzed using an LSRFortessa cell analyzer and FlowJo software.

**Quantitative RT-PCR**. For analyzing the effect of β-adrenergic stimulation on gene expression, sorted MDSCs were incubated overnight (16 h) with isoproterenol (1 μM) diluted in RPMI 1640 medium supplemented with heat inactivated 10% fetal calf serum, L-glutamine, sodium pyruvate, and Penicillin–Streptomycin. Then the cells were lysed in TRI-Reagent (Sigma) and stored at $-80\,°C$ overnight. Total RNA was isolated according to the protocol supplied with the TRI-Reagent. Total RNA (0.1 μg) was reverse transcribed (RT) using the High-Capacity cDNA Reverse Transcription Kit (Applied Biosystems). Real-time PCR analysis was performed using an Applied Biosystems StepOnePlus Real Time PCR System (Foster City, CA) in two independent experiments in duplicates using the Fast SYBR Green Master Mix (Applied Biosystems). Dissociation analysis was performed at the end of each run to confirm the specificity of the reaction. Real-time PCR efficiencies were determined for all sets of primers used. The cycle conditions for real-time PCR were 95 °C for 20 s, followed by 40 cycles of 95 °C for 3 s, and 60 °C for 30 s, and a melt curve stage (95 °C 15 s, 60 °C 1 min, 95 °C 15 s). Quantification of relative gene expression was performed according to the ΔΔ-CT method[57] using StepOne Software 2.3 (Applied Biosystems), and expressed as fold difference ± s.e. m. For analyzing the effect of β-adrenergic stimulation on gene expression, the same samples were used to monitor expression changes of five genes (VEGF, iNOS, TGFβ, IL-10, and TNFα). For analyzing the effects of VTA-activation on gene expression, the same samples were used to monitor the expression changes of the gene CD31. To this end, we performed the Bonferroni multiple testing correction with a cutoff *P*-value of 0.1.

The following primers were used:
GAPDH forward: 5′- TGAAGCAGGCATCTGAGGG-3′,
GAPDH reverse: 5′- CGAAGGTGGAAGAGTGGGAG-3′,
ADRB2 forward 5′- TGGTTGGGGCTACGTCAACTC-3′,
ADRB2 reverse: 5′- CCAGCTGACAAGTGTTTGGC-3′
VEGF forward: 5′- GGAGATCCTTCGAGGAGCACTT-3′,
VEGF reverse: 5′- GGCGATTTAGCAGCAGATATAAGAA-3′
iNOS forward: 5′- GTTCTCAGCCCAACAATACAAGA-3′,
iNOS reverse: 5′- GTGGACGGGTCGATGTCAC-3′
TGFβ forward: 5′- AAACGGAAGCGCATCGAA-3′,
TGFβ reverse: 5′- GGGACTGGCGAGCCTTAGTT-3′
IL-10 forward: 5′- GCTCTTACTGACTGGCATGAG-3′,
IL-10 reverse: 5′- CGCAGCTCTAGGAGCATGTG-3′
TNFα forward: 5′- CCTTTCACTCACTGGCCCAA-3′,
TNFα reverse: 5′- AGTGCCTCTTCTGCCAGTTC-3′
CD31 forward: 5′- TCCCCACCGAAAGCAGTAAT-3′
CD31 reverse: 5′- CCCACGGAGAAGTACTCTGTCTATC-3′

**In vitro T cell suppression assay**. For all in vitro assays, $CD4^+$ T cells were obtained from spleens of naïve mice, and isolated using EasySep mouse $CD4^+$ T cell isolation kit (Stemcell Technologies Inc.). The T cells were labeled with CSFE (Biolegend, San Diego, CA, US) to determine CD4 cell proliferation.

To assess the effects of noradrenergic signaling on bone marrow MDSCs, these cells were isolated from the bone marrow (the site in which MDSCs expressed the highest levels of mRNA for the β2-adrenergic receptor) of tumor bearing mice, by staining and sorting for $CD45^+CD11b^+Gr-1^+$ cells using a FACSaria. Isolated MDSCs ($1 \times 10^5$) were incubated for 1 h in 5 μM of isoproterenol (Sigma–Aldrich, St. Louis, MO, USA) diluted in RPMI media with 1% heat inactivated FBS (Biological Industries, IL). The cells were then washed three times with fresh RMPI medium, resuspended with the CFSE labeled $CD4^+$ T cells ($1 \times 10^5$), which were stimulated with 1 μg/ml of anti-CD3 and anti-CD28 antibodies (Biolegend, San Diego, CA, US) diluted in RPMI medium supplemented with 10% fetal bovine

serum (FBS; Biological Industries, IL), 2 mM L-glutamine (Biological Industries, IL), 1 mM sodium pyruvate (Biological Industries, IL), and 1% PenStrep (Biological Industries, IL), and cultured in a U-shaped 96-well dish. After 96 h, T cells were stained with APC-conjugated anti-CD4 (GK1.5) and analyzed with an LSRFortessa cell analyzer and FlowJo software.

To evaluate the ability of MDSCs to suppress T-cell proliferation, we isolated MDSCs from the tumors because in this site MDSCs manifest the highest immunosuppressive capacity. The MDSCs were isolated from VTA-activated mice and their controls (injected with a virus lacking the information encoding for DREADDs). The cells were then stained for CD45$^+$CD11b$^+$Gr-1$^+$ and sorted using FACSaria. The isolated MDSCs (1×10$^5$) were resuspended in fresh RMPI media (supplemented with 10% fetal bovine serum (FBS; Biological Industries, IL), 2 mM L-glutamine (Biological Industries, IL), 1 mM sodium pyruvate (Biological Industries, IL), and 1% PenStrep (Biological Industries, IL)), and plated in a U-shaped 96-well culture plates, along with CD4$^+$ T cells (1×10$^5$). The T cells used for the incubation were labeled with CFSE and stimulated with 1 µg/ml of anti-CD3 and anti-CD28 antibodies (Biolegend, San Diego, CA, US). Following 96 h, we stained the T cells with an APC-conjugated anti-CD4 (GK1.5) and analyzed the cells with LSRFortessa cell analyzer. We identified proliferating CD4 T cells based on their CFSE levels, which is reduced in proliferating cells. The percentage suppression of proliferating CD4 T cells was calculated using the following equation:

$$\% \text{ Suppression} = 100 - \frac{\text{Proliferating CD4 T cells co−cultured with MDSCs (from control or VTA −activated mice)}}{\text{Anti−CD3+anti−CD28 stimulated CD4 T cells in the absence of MDSCs}}$$

**CD11b$^+$Gr-1$^+$ MSDCs adoptive transfer assay**. CD11b$^+$GR1$^+$ cells were isolated using FACSaria from the tumors of VTA-activated mice and their controls (injected with a virus lacking the information encoding for DREADDs). Isolated MDSCs (2×10$^5$) were co-injected s.c with 2×10$^5$ LLC cells, into naïve C57BL/6 mice. To physically bind the isolated MDSCs and LLC cells, these cells were injected together with 200 µl of Cultrex gel (Trevigen, Gaithersburg, Maryland, USA). Tumor weight was measured after 14 days.

**Statistical analysis**. Significance levels of other data were determined by using Prism5 (GraphPad Software). Experiments were analyzed by two-tailed Student's $t$-test or by Mann–Whitney test to account for the multiple comparisons, as indicated for each experiment.

**Data availability**. Data are available on request from the authors.

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

## Acknowledgements

We would like to thank Z. Ronai, N. Karin, G. Wildbaum, H. Razon, and S. Avraham for helpful discussions, and Y. Shaked for helpful advice and the B16 cell line; Y. Ofran for the Lewis Lung Carcinoma cell line and A. Aronheim for sharing the GFP mice. We would like to thank S. Schwarzbaum for editing the manuscript. We are grateful to O. Shenker, A. Grau, E. Suss-Toby, Y. Sakoury, and M. Holdengreber from the Biomedical Core Facility at the Technion Faculty of Medicine for technical support. This study was supported by the FP-7-CIG grant 618654 (A.R.), the Israel Science Foundation (ISF) grants 1862/15 (A.R.), the Adelis Foundation (A.R.), AICR (F.H. and A.R.), Colleck Research fund, and the Allen and Jewel Prince Center for Neurodegenerative Processes of the Brain. A.R. is The Howard Hughes Medical Institute (HHMI)-Wellcome Trust International scholar.

## Author contributions

T.L.B.-S. and M.S. designed and carried out all the experiments, interpreted the results, and wrote the manuscript; H.A.-D. and B.K. contributed to the experimental design and execution of the experiments, contributed to data analysis and to the manuscript; N.B., T.K., and M.K. contributed to the execution of the experiments, to the interpretation of the results, and the manuscript; J.S. and M.A.R. contributed to the interpretation of the results and to the manuscript; F.H. and A.R. conceived the project, contributed to the experimental design and the interpretation of results, and wrote the manuscript.

## Additional information

**Competing interests:** The authors declare no competing interests.

