## [Peer Review File · Nature Communications]

Reviewers' comments:

Reviewer #1 (Remarks to the Author):

This work by Ben-Shaanan and Schiller et al. follows up on their recently published Nature Medicine paper in which they reported that VTA activation can alter peripheral immune responses and contribute to improved pathogen control. For this study, they wanted to address whether VTA activation similarly promotes enhanced anti-tumor immune responses. They report here that VTA activation does indeed lead to improved tumor control and that this is mediated at least in part through modulation of CD11b+Gr-1+ cell activity. They also employ pharmacological approaches to support a role for the sympathetic nervous system (SNS) and noradrenergic input in VTA activation associated boosting of anti-tumor immune responses. This area of research is compelling and timely and the Rolls lab is uniquely positioned to execute these studies. However, there are several key issues, which I believe need to be addressed in order to solidify their central findings and conclusions.

Major comments:

- (1) It would be helpful in the evaluation of their findings to know whether VTA activation influences the efficiency of cellular adoptive transfer. For instance, does their treatment conditions promote a more robust peritoneal immune response that results in an initial rejection of tumor cells and/or MDSCs immediately following injection?
- (2) Control images and quantification are needed to validate equal neuronal infection between control and VTA virus injections in Figure 1B-C and Supplementary Figures 1-2.
- (3) Control groups should be more extensively described in all of the figure legends. For instance, are the control groups AAV8-hSyn-DIO-mCherry infection plus CNO treatment? Or just one or the other? This should be stated in each of the figure legends.
- (4) Confirmation is needed to validate that DREADD-induced activation of cells in the VTA is specifically responsible for orchestrating the observed changes in anti-tumor immune responses. It could be that TH+ cells in other regions of the brain are also infected. To demonstrate that the reported effects are due to DREADD-mediated activation only in the VTA, the authors should additionally evaluate mCherry staining in other brain regions that have also been proposed to wire the brain's reward system, including the nucleus accumbens, lateral hypothalamus frontal cortex, etc.
- (5) Utilization of 6OHDA to ablate the SNS could potentially have off-target effects that may influence the interpretation of their data. Therefore, a secondary approach to ablate SNS function would help to strengthen their conclusion that the SNS is mechanistically involved in mediating the effects of VTA activation on tumor eradication. Furthermore, the authors should also confirm that 6OHDA does not cross the BBB in their model. They cite that 6OHDA does not normally cross the BBB under homeostatic conditions. However, they are now introducing a number of variables into their system including tumor-induced inflammation and CNO treatment, both of which could potentially influence the integrity of the BBB separately or in tandem.
- (6) The authors rely on Gr-1+CD11b+ staining to identify MDSCs in their studies. The possibility exists that some of their differences that they ascribe to MDSCs in VTA-activated vs. control mice are actually due to the effects of VTA stimulation on neutrophil responses.

Gr-1 stains both Ly6C and Ly6G, thus their gating strategy could include both neutrophils (CD11b+Ly6G+) and MDSCs (CD11b+Ly6C/G+).

(7) The induction of p-CREB MFI levels following beta-agonist treatment is not all that striking. Representative histogram plots should be provided to accompany the MFI data plotted in Figure 3C to properly evaluate these differences.

(8) The technical aspects of their MDSC:T cell suppression experiments are not adequately described in the methods, figure legend, or actual text. It is unclear if they are measuring percentage of total T cells that have proliferated, accumulation of proliferating T cells, or something else. Moreover, with in vitro suppression assays it is usually good practice to provide a dose curve with decreasing ratios of the suppressive cell population to account for culture related issues and other variables.

(9) Only evaluating CD69 expression by tumor-infiltrating T cells is not sufficient to report an effect of VTA activation on in vivo anti-tumor T cell responses. CD69 does not really indicate much in terms of anti-tumor effector T cell function as it can be up regulated by antigen exposure or type I IFNs. To corroborate that VTA activation attenuates the immunosuppressive activities of MDSCs on in vivo anti-tumor T cell responses, the authors should evaluate T cell-mediated cytokine production and/or cytolytic function.

(10) The results presented in Figure 4 are very compelling. However, these findings also raise a number of important questions about MDSC plasticity and longevity. For instance, it is unclear if the MDSCs are completely reprogrammed by the VTA activation and are no longer able to revert back to a pro-tumor phenotype. To have such a profound effect over 14 days would suggest that MDSCs survive a long period of time in vivo and do not turnover. How long do the adoptively transferred MDSCs survive in vivo? It is unclear if long-term survival of MDSCs is required for their effect or if only a short period of time is needed for them to influence the anti-tumor T cell response. It could also be that they are transferring over more neutrophils in the VTA-activated group (see above comment # 6). Or conversely, it could be that VTA-activated MDSCs do not survive as well following adoptive transfer. Understanding what contributes to this effect would be valuable to know in order to better appreciate what biologically underlies these results. Moreover, it would also be important to know whether these differences in tumor load are associated with altered anti-tumor T cell responses or if this occurs independently.

Minor comments

(1) It is unclear how NA levels were measured in Figure 2? qRT-PCR?

Reviewer #2 (Remarks to the Author):

In this manuscript, Ben-Shaan et al expand on their interesting observation (published in Nat. Med) that activation of the reward system can lead to the stimulation of systemic immunity. Here, they demonstrate that DREADD expression in VTA can abrogate the growth of two subcutaneous tumor models (LLC lung cancer and B16 melanoma). Finding reported here are interesting and consistent with their previous observations. However, several issues have to be addressed before the manuscript is accepted for publication.

The exact mechanism by which VTA activation inhibited tumor growth still remains unclear.

Although, they suggest modulation of the bone marrow MDSCs by sympathetic nervous system (SNS) to be important, other potential mechanisms (such as activation of other innate immune cells such as NK cells) have not been explored. Furthermore, other microenvironmental changes (such as angiogenesis) were not examined. Additional mechanistic studies (highlighted below) would have strengthened this paper.z

Fig. 1: Durability of VTA activation (as measured by c-Fos expression) is only presented at 14 days post tumor implantation. Because some experiments extended to 21 and 28 days (Fig. 1g and 1h), persistence of VTA activation should be confirmed in these groups as well. Additionally, have they tested the impact of DREADD expression in other CNS locations besides VTA? Could their observations be due to generalized CNS DREADD expression or is it unique to VTA.

Fig. 2: Tumors in 6OHDA-treated mice (irrespective of VTA activation) appear to be twice as large as the untreated mice (Fig. 2b vs. 1f). Could 6OHDA directly stimulate tumor growth? If so, does this occur through inhibition of the SNS? Or, does 6OHDA have a direct impact on immune cells that express NA receptors? How about angiogenesis?

Fig. 3: TNF α expression was significantly increased in the spleen of VTA-activated mice (Fig. 3h) yet NA levels were not changed (Fig. 1e). This observation appears to be inconsistent with the proposed mechanism of SNS inactivation of MDSCs? Have the authors examined the activity of NK cells (or other CD3 cells) in the spleen or in the tumors in their model? Characterization of MDSC activity also appears to be superficial. Additional techniques, such as flow (to examine the MDSC or TAM phenotype markers like Arg, IDO, etc) or RNA-seq (nanotstring) would have provided more comprehensive information about MDSC polarization in VTA-activated mice.

Fig. 4: Have the authors performed depletion studies (Gr1 Ab) to confirm the role of MDSCs in this model?

Reviewer #3 (Remarks to the Author):

This is a very exciting set of experimental results regarding the CNS pathways that regulate the effects of peripheral neurobiology on tumor growth. The core model supported by these studies is that VTA activation suppresses peripheral SNS signaling (particularly in bone marrow) and thereby reduces the production of MDSCs that would otherwise subsequently inhibit cellular immune responses against cancer. The authors present a promising set of studies to support this hypothesis, but several additional highly feasible studies would be required to more definitively support it. Please consider the remarks below as encouraging suggestions to enhance the impact of this already promising work.

Abstract:

Needs to include a description of tumor model/s used as a read-out (i.e., LLC and B16).

Abstract lacked any allusion to studies blocking MDSCs in the VTA-activation model in order to fully establish causation. Mimicry of VTA-activation effects in naive animals by adoptive transfer of MDSCs from VTA-activated mice shows that MDSCs are sufficient to facilitate

tumor growth, but does not establish whether they are necessary. For that MDSC inhibition (e.g., by inhibiting myeloid cell growth differentiation in general, or MDSC differentiation in particular) is required.

Introduction:

First 5 refs listed are not particularly comprehensive, and some are controversial/discredited. Perhaps cite more comprehensive review articles (e.g., 2 from Paige Green and colleagues in Nature Reviews Cancer and/or Andrew Steptoe et al in Nat Clin Pract Onc)?

It is not accurate to claim that mechanisms of psychobiological effects on cancer remain “unknown”; quite a lot is now known about peripheral neural, cellular, and molecular mechanisms of such effects (see the 2 review articles referenced above in Nat Rev Cancer). More accurate would be to claim that little is known about the CNS mechanisms involved, where this paper makes a very significant contribution (but see also Cao et al, Cell. 2010 Jul 9;142(1):52-64)

Need to clarify the nature of LLC cells when initially referenced in the text (what kind of cells are these, and how was their identity verified), along with the locale of sc injection.

6OHDA is not a clean abrogation of SNS (it induces massive catecholamine release prior to nerve fiber death). Results would be more persuasive if data show that pharmacologic abrogation of beta-adrenergic signaling (preferably with an agent that does not cross BBB, e.g., nadolol) also blocks effects of VTA activation on tumor growth.

The experiment in Fig 2 (6OHDA) lacks a positive control. Needs to be repeated in parallel with a sham SNS intervention (e.g., saline) that continues to show VTA activation of tumor growth in the same experiment as it is abrogated by 6OHDA (i.e., a 2 x 2 design instead of the current 2 condition experiment that attempts to compare with other separate experiments).

“Stress responses” can occur in the bone marrow, so it is inaccurate to equate stress with plasma catecholamine levels (as the ms appears to do now). More accurate to simply state no VTA activation-induced difference observed in plasma but significant difference observed in bone marrow.

SNS directly innervates some tumors, but certainly not all. Unless authors directly document SNS innervation of all/most tumors in the particular model examined here (e.g., by histological detection of nerve fibers in tumor tissue as in Fig 2h for bone marrow), it would be more accurate to say SNS innervates some tumors, or that SNS can innervate tumors, or something similarly less general. The intratumoral catecholamines assessed here may come predominately from blood or perivascular nerve fibers rather than from true innervation of the tumor parenchyma per se.

The results in Fig 3 are interpreted as showing that VTA activation suppresses SNS output

to the bone marrow and thereby inhibits the suppressive effect of MDSCs on T cell-mediated responses against cancer. However, the final step is not directly tested. One simple way to do that would be to conduct the VTA activation protocol in SCID or nude mice that lack functional T cells (in parallel with their parental strains bearing functional T cells).

Upregulated TNF production is implied to mediate VTA/SNS effects on T cell proliferation/activation, but that too is not directly demonstrated. Experiments with TNF knockout mice or anti-TNF neutralizing antibody would support that claim.

According to the experimental schematic in Fig 4, adoptive transfer of PMN-MDSCs into naïve mice is sufficient to mimic effects observed in Fig 1. Several questions arise, including whether similar effects occur with adoptive transfer M-MDSCs?, whether comprehensive numerical measures of tumor size are also affected (e.g., as in Fig 1e)?, and whether inhibition of MDSCs in the VTA-activated mice blocks effects on tumor growth in those animals.

Much is known about how SNS activation in bone marrow enhances myeloid differentiation (e.g., from studies by P Frenette et al., J Sheridan's group and particularly a study by Powell et al, and the group of Nahrendorf & Swirski). That material should be cited and could help design experiments identifying the molecular mechanism by which VTA-induced SNS downregulation inhibits MDSC development/distribution. One approach might involve analysis of the myeloid growth factors already demonstrated to be upregulated by beta-adrenergic signaling (e.g., as in Sheridan's and Frenette's studies).

The Discussion needs to acknowledge some limitations on the scope of conclusions that can be drawn from these studies. E.g., LLC and B16 are poor models of tumor initiation or metastasis; they assess predominately primary tumor growth rates in already-established cancer cells. There is no verification of relevance to human cancer, or to tumor growth in orthotopic tissue settings. Tumor injections into blood sc are not broadly representative of normal conditions where tumors initiate in solid tissues. The number of cell lines examined is limited, and the present models may overestimate the breadth of effects on other tumor types that are not highly immunogenic or lung localized. All that said, the authors are to be complimented for their careful interpretation of these studies as a test of basic physiological relationships among the nervous, immune, and tumor systems, as opposed to claiming a mechanism for the controversial clinical relationships between positive mood and cancer progression.

May 8, 2018

We would like to thank the referees for their time and attention in reviewing our manuscript entitled "Modulation
of anti-tumor immunity by the brain's reward system".

In this manuscript, we provided the first demonstration that reward system activity can alter the anti-tumor
immune response. Thanks to the reviewers' comments, we clarified the manuscript and performed several
additional key experiments that substantiated our findings and expanded our understanding of the mechanisms
underlying this effect. Below is a summary of these main new experiments:

1. To demonstrate the involvement of the sympathetic nervous system in mediating reward system effects, we
originally used 6OHDA. We now added another experiment using the β -adrenergic blocker, Nadolol (Fig 2b).
This manipulation, similarly to the effect of 6OHDA, eliminated the beneficial effects of the reward system on
tumor growth.

2. We originally applied an adoptive transfer experiment to demonstrate that MDSCs were sufficient to mediate
the VTA-induced attenuation in tumor growth. We now added an experiment using anti-Gr-1 antibody to deplete
MDSCs in VTA activated mice, demonstrating that MDSCs are also necessary for the beneficial effect of VTA on
tumor growth (Fig 4b).

3. Originally, we demonstrated that MDSCs transferred to recipient mice were sufficient to attenuate tumor
growth, but we did not verify that these cells survive within the recipient mice. To address this point, we repeated
this transfer experiment using GFP mice as recipients. We analyzed the recipients at three different time points
(Fig S14) following transplantation, and we now show that the MDSCs survive in the recipient mice.

4. We provide new evidence that in the VTA-activated mice, CD8 T cells demonstrated elevated levels of
Granzyme B (Fig 4a). Moreover, this effect appears to depend on MDSCs, because depletion of MDSCs
eliminated the effect of VTA activation on Granzyme B levels.

In addition, as detailed in the point by point response, we address all the specific concerns raised by the
reviewers.

Reviewers' comments:

Reviewer #1 (Remarks to the Author):

***This work by Ben-Shaanan and Schiller et al. follows up on their recently published Nature Medicine***
***paper in which they reported that VTA activation can alter peripheral immune responses and contribute***
***to improved pathogen control. For this study, they wanted to address whether VTA activation similarly***
***promotes enhanced anti-tumor immune responses. They report here that VTA activation does indeed***
***lead to improved tumor control and that this is mediated at least in part through modulation of CD11b+Gr-***
***1+ cell activity. They also employ pharmacological approaches to support a role for the sympathetic***
***nervous system (SNS) and noradrenergic input in VTA activation associated boosting of anti-tumor***
***immune responses. This area of research is compelling and timely and the Rolls lab is uniquely***
***positioned to execute these studies. However, there are several key issues, which I believe need to be***
***addressed in order to solidify their central findings and conclusions.***

We would like to thank the reviewer for the evaluation of the work and the helpful suggestions.

***Major comments: Please note that we merged points 1 and 10 as they both relate to the transfer***
***experiment.***

**(1) It would be helpful in the evaluation of their findings to know whether VTA activation influences the**
**efficiency of cellular adoptive transfer. For instance, does their treatment conditions promote a more**
**robust peritoneal immune response that results in an initial rejection of tumor cells and/or MDSCs**
**immediately following injection?**

**(10) The results presented in Figure 4 are very compelling. However, these findings also raise a number**
**of important questions about MDSC plasticity and longevity. For instance, it is unclear if the MDSCs are**
**completely reprogrammed by the VTA activation and are no longer able to revert back to a pro-tumor**
**phenotype. To have such a profound effect over 14 days would suggest that MDSCs survive a long period**
**of time in vivo and do not turnover. How long do the adoptively transferred MDSCs survive in vivo? It is**
**unclear if long-term survival of MDSCs is required for their effect or if only a short period of time is**
**needed for them to influence the anti-tumor T cell response. It could also be that they are transferring**
**over more neutrophils in the VTA-activated group (see above comment # 6). Or conversely, it could be**
**that VTA-activated MDSCs do not survive as well following adoptive transfer. Understanding what**
**contributes to this effect would be valuable to know in order to better**
**appreciate what biologically underlies these results. Moreover, it would also be important to know**
**whether these differences in tumor load are associated with altered anti-tumor T cell responses or if this**
**occurs independently.**

The reviewer raises several important questions regarding the nature of the adoptive transfer. Below we have
summarized and answered these concerns.

1) Can the reduction in tumor size be explained by initial elimination of the tumor cells?

The transfer experiment was originally designed to determine whether the effect of VTA activation on MDSCs is
sufficient to account for the reduction in tumor weight. To this end, we injected MDSCs along with an equal
number of tumor cells, and evaluated the effects on tumor size 14 days following the transfer. To determine
whether the effect occurs immediately after transplantation (e.g. initial rejection of the tumor cells) or if it requires
time to develop, we now added another experiment, designed to evaluate the effect on tumor growth at an earlier
time point, 7 days after the transfer. Our findings suggest that there wasn't an initial rejection of the tumor cells
in the VTA-activated group, as there was no difference in the tumor weight at this time point (Fig S15; $p < 0.2417$).

2) Do transferred MDSCs survive in the recipient mice?

We determined whether and for how long the transferred MDSCs can survive in the tumor environment of the
recipient mice. We transferred MDSCs along with tumor cells, into GFP mice. The expression of GFP by the
recipient mice allowed us to distinguish between the transferred and resident MDSCs populations. We analyzed
the survival rate of the transferred MDSCs at three different time points following transplantation: 4, 7 and 14
78 days. As shown in Figure S14, transplanted MDSCs represented $37.1 \pm 2.7\%$ of total MDSCs 4 days after
79 transplantation, $28.9 \pm 3.8\%$ at 7 days, and $9.1 \pm 1.9\%$ at 14 days. Thus, we demonstrate that MDSCs survive in
the tumor microenvironment, though their abundance is reduced over time.

3) Are the effects on T cells MDSCs-dependent, or do they occur independently?

We extended our characterization of the effector T cell functions and demonstrated that CD8 T cells in VTA
activated mice, manifest an increase in Granzyme B expression. We further show that this increase is MDSCs-
dependent, as MDSCs depletion (using systemic injection of anti Gr-1 antibody), eliminated this difference in
Granzyme B levels by CD8 T cells (Fig 4a).

4) The reviewer correctly indicates that there is no reliable tool to differentiate between neutrophils and PMN-
MDSCs (also referred as granulocytic MDSCs). We now emphasize this point in the text (pages 4,6). However,
to address this issue, at least in part, we now provide evidence that VTA activation did not affect the relative
abundance of M-MDSCs and PMN-MDSCs (Fig. S7).

**(2) Control images and quantification are needed to validate equal neuronal infection between control**
**and VTA virus injections in Figure 1B-C and Supplementary Figures 1-2.**

We now calibrated the percentage of cells infected with the DREADD-encoding virus versus infection with the
control virus. As shown in Fig. S1, there was no significant difference in the percentage of infected cells ($p < 0.26$).

**(3) Control groups should be more extensively described in all of the figure legends. For instance, are**
**the control groups AAV8-hSyn-DIO-mCherry infection plus CNO treatment? Or just one or the other?**
**This should be stated in each of the figure legends.**

Thank you for highlighting this omission. We added full descriptions of the control groups in all figure legends.

**(4) Confirmation is needed to validate that DREADD-induced activation of cells in the VTA is specifically**
**responsible for orchestrating the observed changes in anti-tumor immune responses. It could be that**
**TH+ cells in other regions of the brain are also infected. To demonstrate that the reported effects are due**
**to DREADD-mediated activation only in the VTA, the authors should additionally evaluate mCherry**
**staining in other brain regions that have also been proposed to wire the brain's reward system, including**
**the nucleus accumbens, lateral hypothalamus frontal cortex, etc.**

As suggested by the reviewer, we now provide representative immunofluorescence images of the nucleus
accumbens, lateral hypothalamus, and frontal cortex from the brains of mice injected with a virus (Fig. S2). There
was no mCherry expression in these brain regions, demonstrating that DREADD expression was restricted to
the VTA.

**(5) Utilization of 6OHDA to ablate the SNS could potentially have off-target effects that may influence the**
**interpretation of their data. Therefore, a secondary approach to ablate SNS function would help to**
**strengthen their conclusion that the SNS is mechanistically involved in mediating the effects of VTA**
**activation on tumor eradication. Furthermore, the authors should also confirm that 6OHDA does not**
**cross the BBB in their model. They cite that 6OHDA does not normally cross the BBB under homeostatic**
**conditions. However, they are now introducing a number of variables into their system including tumor-**
**induced inflammation and CNO treatment, both of which could potentially influence the integrity of the**
**BBB separately or in tandem.**

This is an important question. Therefore, as a complementary approach, we used Nadolol, a β -adrenergic
antagonist that does not cross the BBB, an approach suggested by Reviewer #3 to consolidate the involvement
of the SNS in mediating the beneficial effects of VTA activation on tumor weight. In analogy to the effect of
6OHDA, Nadolol treatment eliminated the effects of VTA activation on tumor growth (Fig. 2b). These findings
further support the mechanistic involvement of the sympathetic nervous system in VTA-induced tumor
suppression.

Moreover, to confirm that catecholaminergic neurons in the brain were not affected by the 6OHDA treatment, we
demonstrate that there was no significant difference in the number of TH⁺ cells in the VTA of tumor bearing mice
following 6OHDA or vehicle injection (Fig. S4).

**(6) The authors rely on Gr-1+CD11b+ staining to identify MDSCs in their studies. The possibility exists**
**that some of their differences that they ascribe to MDSCs in VTA-activated vs. control mice are actually**
**due to the effects of VTA stimulation on neutrophil responses. Gr-1 stains both Ly6C and Ly6G, thus**
**their gating strategy could include both neutrophils (CD11b+Ly6G+) and MDSCs (CD11b+Ly6C/G+).**

As the reviewer correctly indicates, based on the literature and in agreement with our gating strategy,
CD11b+Ly6G+ cells may be comprised of both granulocytic MDSCs and neutrophils. Because there is no clear
way to distinguish between these populations, we refer to these cells collectively as PMN-MDSCs (Neutrophils
and polymorphonuclear myeloid-derived suppressor cells). We now discuss this point in the manuscript on pages

4 and 6. In addition, in Figure S7, we demonstrate that there was no difference between the VTA-activated mice
and controls in the relative abundance of PMN-MDSCs identified by their expression of CD11b and Ly6G versus
M-MDSCs identified by their expression of CD11b and Ly6C.

**(7) The induction of p-CREB MFI levels following beta-agonist treatment is not all that striking.**
**Representative histogram plots should be provided to accompany the MFI data plotted in Figure 3C to**
**properly evaluate these differences.**

As suggested by the reviewer, we added a representative histogram showing the change in p-CREB levels
following β -agonist treatment (Fig S6). Moreover, we agree with the reviewer's comment that changes in p-CREB
MFI are not striking. One factor that can explain this phenomenon, is that the cells used for the analysis were
isolated from the bone marrow, a niche with relatively high NA levels (Fig S5). To substantiate this claim, we provide
a reviewer figure, demonstrating a representative staining of bone marrow MDSCs isolated from tumor bearing mice
treated with a β -adrenergic antagonist (Nadolol). Indeed, in Ly6G⁺ and Ly6C⁺ bone marrow cells of Nadolol treated
mice, p-CREB staining was lower than vehicle treated mice, corroborating the capacity of p-CREB staining as a
reporter of β -adrenergic activity.

**(8) The technical aspects of their MDSC: T cell suppression experiments are not adequately described in**
**the methods, figure legend, or actual text. It is unclear if they are measuring percentage of total T cells**
**that have proliferated, accumulation of proliferating T cells, or something else. Moreover, with in vitro**
**suppression assays it is usually good practice to provide a dose curve with decreasing ratios of the**
**suppressive cell population to account for culture related issues and other variables.**

We have now expanded the description of the experiment in Figure legend 3i, 3k and in the Methods (page 25).
As advised by the reviewer, we now provide the relevant controls in Fig S11, including: activated and non-
activated CFSE labelled CD4 T cells cultured without MDSCs, as well as CD4 T cells cultured with MDSCs from
VTA activated mice, and with MDSCs from sham-virus injected mice.

Regarding the use of a dose response: Although a dose curve was applied while calibrating the initial
experimental protocol, due to the limited availability of the MDSCs from VTA activated mice and controls, we had
to limit the experimental design to a single dose.

**(9) Only evaluating CD69 expression by tumor-infiltrating T cells is not sufficient to report an effect of**
**VTA activation on in vivo anti-tumor T cell responses. CD69 does not really indicate much in terms of**
**anti-tumor effector T cell function as it can be up regulated by antigen exposure or type I IFNs. To**
**corroborate that VTA activation attenuates the immunosuppressive activities of MDSCs on in vivo anti-**
**tumor T cell responses, the authors should evaluate T cell-mediated cytokine production and/or cytolytic**
**function.**

The reviewer correctly states that changes in CD69 expression do not necessarily represent a functional change
in an effector cell. To directly address the reviewer's comment, we further characterized tumor and spleen CD4
and CD8 T cell abundance, IFN- γ and TNF- α expression, as well as Granzyme B expression by tumor CD8 T
cells. We did not observe any difference in the abundance of the T cells or their TNF- α /IFN- γ levels (Fig S12).
However, we did find an increase in Granzyme B levels in tumor-associated CD8 T cells of VTA-activated mice
compared to controls ($p < 0.0159$; Fig 4a, Fig S13). This further supports our findings that VTA activations induces
anti-tumor immune response.

**Minor comments**

**(1) It is unclear how NA levels were measured in Figure 2? qRT-PCR?**

Levels of NA were measured by ELISA and validated by immunohistochemistry. We now clarify this point in the
legend of Figure 2.

**Reviewer #2 (Remarks to the Author):**

*In this manuscript, Ben-Shaan et al expand on their interesting observation (published in Nat. Med)*
*that activation of the reward system can lead to the stimulation of systemic immunity. Here, they*
*demonstrate that DREADD expression in VTA can abrogate the growth of two subcutaneous tumor*
*models (LLC lung cancer and B16 melanoma). Finding reported here are interesting and consistent with*
*their previous observations. However, several issues have to be addressed before the manuscript is*
*accepted for publication. The exact mechanism by which VTA activation inhibited tumor growth still*
*remains unclear. Although, they suggest modulation of the bone marrow MDSCs by sympathetic nervous*
*system (SNS) to be important, other potential mechanisms (such as activation of other innate immune*
*cells such as NK cells) have not been explored. Furthermore, other microenvironmental changes (such*
*as angiogenesis) were not examined. Additional mechanistic studies (highlighted below) would have*
*strengthen this paper.*

We would like to thank the reviewer for the helpful comments and suggestions.

**1. Fig. 1: Durability of VTA activation (as measured by c-Fos expression) is only presented at 14 days**
**post tumor implantation. Because some experiments extended to 21 and 28 days (Fig. 1g and 1h),**
**persistence of VTA activation should be confirmed in these groups as well. Additionally, have they**
**tested the impact of DREADD expression in other CNS locations besides VTA? Could their**
**observations be due to generalized CNS DREADD expression or is it unique to VTA.**

To address the reviewer's comment, we now provide c-Fos calibration on day 28. Our results demonstrate that
DREADD manipulation was also effective at this time point (Fig S3; $p < 0.012$).

The reviewer also correctly suggests that activity of other brain regions may also affect tumor growth. In fact, a
major focus of my lab is to characterize the peripheral immune changes induced by activity of different brain
regions, as they serve different roles in an organism's behavior and physiology. However, as each of these brain
areas requires extensive characterization, we consider such a survey to be beyond the scope of the current
study.

A third point raised by the reviewer is that the observed effects could be due to generalized CNS DREADD
expression. To address this comment, we provide in Figure S2, an immunohistochemical analysis of mCherry
labeling (a marker of DREADD expression in our system) to demonstrate that there is no neuronal expression of
mCherry beyond the VTA dopaminergic neurons.

**2. Fig. 2: Tumors in 6OHDA-treated mice (irrespective of VTA activation) appear to be twice as large as**
**the untreated mice (Fig. 2b vs. 1f). Could 6OHDA directly stimulate tumor growth? If so, does this occur**
**through inhibition of the SNS? Or, does 6OHDA have a direct impact on immune cells that express NA**
**receptors? How about angiogenesis?**

As the reviewer correctly points out, the tumors in the 6OHDA-treated mice are larger than in un-treated controls.
This is in line with the literature indicating that sympathetic activity plays a role in tumor growth (e.g. L. Horvathova
et al, 2016 and SW. Cole et al, 2015). In this study, we used 6OHDA to eliminate peripheral sympathetic activity
in order to determine whether the SNS is required to mediate the signal between the brain's reward system to
the periphery. We found that in the absence of an active SNS, the VTA had no effect on tumor growth. In the

revised version, we took an additional approach to validate SNS involvement in mediating the effects of the VTA,
which was suggested by Reviewer #3. Instead of 6OHDA, we treated VTA activated mice and their controls with
Nadolol, a β -adrenergic blocker. Similar to the effect of 6OHDA, we found that Nadolol treatment abrogated VTA
effects on tumor growth (Fig 2b), further demonstrating SNS involvement in VTA-induced reduction in tumor
weight.

Regarding the potential effects of VTA activation on angiogenesis, we could not find any differences in MDSCs
RNA expression of VEGF following exposure to a β -adrenergic agonist or from VTA-activated mice and their
controls (Fig S8, S9). Similarly, there was no significant difference in mRNA levels of CD31 in total tumor tissue
following VTA activation (Fig S10).

**3. Fig. 3: TNF α expression was significantly increased in the spleen of VTA-activated mice (Fig. 3h) yet**
**NA levels were not changed (Fig. 1e). This observation appears to be inconsistent with the proposed**
**mechanism of SNS inactivation of MDSCs? Have the authors examined the activity of NK cells (or other**
**CD3 cells) in the spleen or in the tumors in their model?**

Indeed, as noted by the reviewer, NA levels were not affected in the spleen, tumor, or plasma (Fig 2d-f). However,
in the bone marrow, the production site of MDSCs, NA levels were reduced in the VTA-activated mice (Fig 2g-i).
Thus, this change in the milieu where the MDSCs develop, may affect their subsequent functional profile. We
now discuss this point on page 5.

Addressing the reviewer's suggestion that we characterize additional CD3 cell populations, we found that CD8
cells from VTA-activated mice expressed elevated Granzyme B levels (Fig 4a, Fig S13), which is consistent with
the reduction in tumor size. Thus, we did not expand our analysis to NK, although, based on an initial screen we
performed, we did not observe any effect on their expression of Granzyme B (tumor $p < 0.1425$, spleen $p < 0.6051$),
IFN- γ (tumor $p < 0.2514$, spleen $p < 0.6873$) or CD27 (tumor $p < 0.6044$, $p < 0.8120$).

**4. Characterization of MDSC activity also appears to be superficial. Additional techniques, such as flow**
**(to examine the MDSC or TAM phenotype markers like Arg, IDO, etc) or RNA-seq (nanosttring) would have**
**provided more comprehensive information about MDSC polarization in VTA-activated mice.**

We now extended our analysis of tumor MDSCs from VTA activated mice and their controls, to include flow
analysis of IDO, Arginase, PD-1, IFN γ and iNOS. None of these markers was altered by VTA activation, as shown
in Figure S9.

**5. Fig. 4: Have the authors performed depletion studies (Gr1 Ab) to confirm the role of MDSCs in this**
**model?**

We would like to thank the reviewer for this important suggestion. We added a demonstration that Gr-1-mediated
depletion abrogated the VTA-induced effects on tumor growth (Fig 4b). Thus, we now show that MDSCs are both
necessary and sufficient to mediate VTA-induced effects on tumor growth.

**Reviewer #3 (Remarks to the Author):**

***This is a very exciting set of experimental results regarding the CNS pathways that regulate the effects***
***of peripheral neurobiology on tumor growth. The core model supported by these studies is that VTA***
***activation suppresses peripheral SNS signaling (particularly in bone marrow) and thereby reduces the***
***production of MDSCs that would otherwise subsequently inhibit cellular immune responses against***
***cancer. The authors present a promising set of studies to support this hypothesis, but several additional***

**highly feasible studies would be required to more definitively support it. Please consider the remarks**
**below as encouraging suggestions to enhance the impact of this already promising work.**

We would like to thank the reviewer for the careful evaluation of the study and the helpful comments.

**1. Abstract needs to include a description of tumor model/s used as a read-out (i.e., LLC and B16).**

We added a description of the relevant cancer models to the abstract.

**2. Abstract lacked any allusion to studies blocking MDSCs in the VTA-activation model in order to fully**
**establish causation. Mimicry of VTA-activation effects in naive animals by adoptive transfer of MDSCs**
**from VTA-activated mice shows that MDSCs are sufficient to facilitate tumor growth, but does not**
**establish whether they are necessary. For that MDSC inhibition (e.g., by inhibiting myeloid cell growth**
**differentiation in general, or MDSC differentiation in particular) is required.**

The reviewer is correct; in the original version of our manuscript, we only demonstrated that MDSCs are sufficient
to mediate the effects of VTA-activation tumor weight via the transfer experiment. To also establish the
requirement for these cells, we performed Gr-1 depletion. In this experiment, we injected VTA-activated mice
and their controls with anti Gr-1 antibody, in order to deplete MDSCs. We now show in Fig 4b that MDSCs
depletion abrogates the effects of VTA on tumor growth. Thus, MDSCs are both sufficient and necessary to
mediate VTA-effects on the tumor.

**3. Introduction: First 5 refs listed are not particularly comprehensive, and some are**
**controversial/discredited. Perhaps cite more comprehensive review articles (e.g., 2 from Paige Green**
**and colleagues in Nature Reviews Cancer and/or Andrew Steptoe et al in Nat Clin Pract Onc)?**

Thank you for pointing out these issues with the papers that we cited. We now replaced the papers that we
initially referenced, with those suggested by the reviewer.

**4. It is not accurate to claim that mechanisms of psychobiological effects on cancer remain “unknown”;**
**quite a lot is now known about peripheral neural, cellular, and molecular mechanisms of such effects**
**(see the 2 review articles referenced above in Nat Rev Cancer). More accurate would be to claim that little**
**is known about the CNS mechanisms involved, where this paper makes a very significant contribution**
**(but see also Cao et al, Cell. 2010 Jul 9;142(1):52-64)**

We modified the text to read: “However, these studies have yielded inconsistent result, and our understanding of
the central neuronal mechanisms underlying the effect of emotional states on cancer is still limited.” (page 2).

**5. Need to clarify the nature of LLC cells when initially referenced in the text (what kind of cells are these,**
**and how was their identity verified), along with the locale of sc injection.**

This was added to the text as part of a more comprehensive description of the tumor model, both in the main text
(page 2) and in the Methods (page 20).

**6. 6OHDA is not a clean abrogation of SNS (it induces massive catecholamine release prior to nerve fiber**
**death). Results would be more persuasive if data show that pharmacologic abrogation of beta-adrenergic**

**signaling (preferably with an agent that does not cross BBB, e.g., nadolol) also blocks effects of VTA**
**activation on tumor growth.**

We would like to thank the reviewer for the suggestion to use Nadolol. Using this agent, we were able to
demonstrate that VTA activation indeed requires sympathetic activity, specifically, β -adrenergic signaling (Fig
2b).

**7. The experiment in Fig 2 (6OHDA) lacks a positive control. Needs to be repeated in parallel with a sham**
**SNS intervention (e.g., saline) that continues to show VTA activation of tumor growth in the same**
**experiment as it is abrogated by 6OHDA (i.e., a 2 x 2 design instead of the current 2 condition experiment**
**that attempts to compare with other separate experiments).**

This is absolutely correct and we now added the relevant experimental conditions to the revised Fig 2a.

**8. "Stress responses" can occur in the bone marrow, so it is inaccurate to equate stress with plasma**
**catecholamine levels (as the ms appears to do now). More accurate to simply state no VTA activation-**
**induced difference observed in plasma but significant difference observed in bone marrow.**

We corrected the text (page 3).

**9. SNS directly innervates some tumors, but certainly not all. Unless authors directly document SNS**
**innervation of all/most tumors in the particular model examined here (e.g., by histological detection of**
**nerve fibers in tumor tissue as in Fig 2h for bone marrow), it would be more accurate to say SNS**
**innervates some tumors, or that SNS can innervate tumors, or something similarly less general. The**
**intratumoral catecholamines assessed here may come predominately from blood or perivascular nerve**
**fibers rather than from true innervation of the tumor parenchyma per se.**

We modified the text to state: "However, the SNS also directly innervates organs relevant to cancer biology,
including some tumors and all immune organs". (Page 4)

**10. The results in Fig 3 are interpreted as showing that VTA activation suppresses SNS output to the**
**bone marrow and thereby inhibits the suppressive effect of MDSCs on T cell-mediated responses against**
**cancer. However, the final step is not directly tested. One simple way to do that would be to conduct the**
**VTA activation protocol in SCID or nude mice that lack functional T cells (in parallel with their parental**
**strains bearing functional T cells). Upregulated TNF production is implied to mediate VTA/SNS effects**
**on T cell proliferation/activation, but that too is not directly demonstrated. Experiments with TNF**
**knockout mice or anti-TNF neutralizing antibody would support that claim.**

The reviewer is absolutely correct that while we established a causal connection between VTA activation and
MDSCs, we did not demonstrate the connection between MDSCs and T cells, or MDSCs-produced TNF- α and
T cell function. Although it will be compelling to perform VTA-activation in T cell deficient mice, our ability to target
the VTA dopaminergic neurons depends on the use of a transgenic mouse strain (TH-Cre). Thus, we cannot
perform this experiment within a reasonable time frame. Similarly, the use of TNF-KO mice would allow us to
establish the connection between MDSCs-produced TNF- α and the effect on T cells; however, generating TH-
Cre+ - TNF-KO mice will require an extensive breeding program. Moreover, TNF- α is produced by various cell
populations (immune and others), and it will thus be difficult to isolate the MDSCs-dependent effect.

Although we could not perform the suggested experiments, we wanted to further establish the connection
between MDSCs and T cell-mediated responses following VTA activation. We found that following VTA
activation, tumor CD8 T cells increase their Granzyme B expression, and this increase is indeed MDSCs-
dependent. We show in Fig 4a that MDSCs depletion (using systemic injection of anti Gr-1 antibody), eliminated
this difference in Granzyme B levels in CD8 T cells. Nevertheless, our findings do not rule out the possibility that
other processes which are MDSCs independent are also involved. We now discuss this point on page 6.

**11. According to the experimental schematic in Fig 4, adoptive transfer of PMN-MDSCs into naïve mice**
**is sufficient to mimic effects observed in Fig 1. Several questions arise, including whether similar effects**
**occur with adoptive transfer M-MDSCs?, whether comprehensive numerical measures of tumor size are**
**also affected (e.g., as in Fig 1e)?, and whether inhibition of MDSCs in the VTA-activated mice blocks**
**effects on tumor growth in those animals.**

We would like to thank the reviewer for pointing out we did not clarify in the original manuscript that we used both
M-MDSCs and PMN-MDSCs. We used unfractionated MDSCs in our transfer experiments, mostly for technical
reasons. It is extremely difficult to purify a sufficient number of M-MDSCs to achieve effective transfer. Therefore,
for the transfer experiment, we isolated the MDSCs based on their expression of Gr-1 and CD11b. We now clarify
this issue on page 6. Nevertheless, we could partially address the question of MDSC subtypes, by demonstrating
that VTA-activation did not alter the relative proportion of PMN-MDSCs and M-MDSCs used for the transfer (Fig.
S7).

The additional numerical measurement of the tumor size, which were suggested by the reviewer, are now
provided in Fig. 4f.

Finally, to determine whether MDSCs inhibition in VTA-activated or control mice affected tumor growth, we
performed the Gr-1-depletion experiment, showing that MDSCs are required to mediate the effect of VTA on
tumor growth. It is now included as Fig. 4b.

**12. Much is known about how SNS activation in bone marrow enhances myeloid differentiation (e.g., from**
**studies by P Frenette et al., J Sheridan's group and particularly a study by Powell et al, and the group of**
**Nahrendorf & Swirski). That material should be cited and could help design experiments identifying the**
**molecular mechanism by which VTA-induced SNS downregulation inhibits MDSC**
**development/distribution. One approach might involve analysis of the myeloid growth factors already**
**demonstrated to be upregulated by beta-adrenergic signaling (e.g., as in Sheridan's and Frenette's**
**studies).**

The reviewer is correct that understanding the underlining molecular mechanism will be beneficial. However,
following consultation with the Editor, we consider the specific molecular characterization will be beyond the
scope of the present study. We discuss this limitation on page 7 and added the proposed citations.

**13. The Discussion needs to acknowledge some limitations on the scope of conclusions that can be**
**drawn from these studies. E.g., LLC and B16 are poor models of tumor initiation or metastasis; they**
**assess predominately primary tumor growth rates in already-established cancer cells. There is no**
**verification of relevance to human cancer, or to tumor growth in orthotopic tissue settings. Tumor**
**injections into blood sc are not broadly representative of normal conditions where tumors initiate in solid**
**tissues. The number of cell lines examined is limited, and the present models may overestimate the**
**breadth of effects on other tumor types that are not highly immunogenic or lung localized. All that said,**
**the authors are to be complimented for their careful interpretation of these studies as a test of basic**
**physiological relationships among the nervous, immune, and tumor systems, as opposed to claiming a**
**mechanism for the controversial clinical relationships between positive mood and**
**cancer progression.**

We agree with the reviewer that the present study serves mainly as an initial indication of the potential
of brain activity, and specifically the reward system, to affect tumor growth. Accordingly, we now expanded the
description of the relevant limitations of our results and modified the text to state: "It is important to note that we
do not consider such robust and specific modulations of VTA activity (as induced here by DREADD activation)
to be physiological and the relevance to human cancer is still unknown. Moreover, in this study, we used two
tumor models that evaluate a specific aspect in tumor biology, namely, primary tumor growth rates. Thus, other
aspects of cancer development and progression such as tumor initiation or metastasis remain unknown. Rather
than dissecting the specific molecular mechanisms mediating these effects, our study aims to demonstrate a

functional connection between a mood-regulating neuronal circuit, such as the reward system, and tumor
growth.”

REVIEWERS' COMMENTS:

Reviewer #1 (Remarks to the Author):

The authors did an outstanding job of addressing my concerns and I believe that the revised manuscript is now suitable for publication at Nature Communications. Congratulations on this excellent work.

Reviewer #3 (Remarks to the Author):

The authors have done an outstanding job of addressing all of the major issues highlighted in my previous review.